# Model-guided development of an evolutionarily stable yeast chassis

Filipa Pereira[1,2,†] (iD), Helder Lopes[3,†] (iD), Paulo Maia[4] (iD), Britta Meyer[5], Justyna Nocon[1], Paula Jouhten[1], Dimitrios Konstantinidis[1] (iD), Eleni Kafkia[1,6] (iD), Miguel Rocha[3], Peter Kötter[5], Isabel Rocha[3,7,*] (iD) & Kiran R Patil[1,6,**] (iD)

## Abstract

First-principle metabolic modelling holds potential for designing microbial chassis that are resilient against phenotype reversal due to adaptive mutations. Yet, the theory of model-based chassis design has rarely been put to rigorous experimental test. Here, we report the development of *Saccharomyces cerevisiae* chassis strains for dicarboxylic acid production using genome-scale metabolic modelling. The chassis strains, albeit geared for higher flux towards succinate, fumarate and malate, do not appreciably secrete these metabolites. As predicted by the model, introducing product-specific TCA cycle disruptions resulted in the secretion of the corresponding acid. Adaptive laboratory evolution further improved production of succinate and fumarate, demonstrating the evolutionary robustness of the engineered cells. In the case of malate, multi-omics analysis revealed a flux bypass at peroxisomal malate dehydrogenase that was missing in the yeast metabolic model. In all three cases, flux balance analysis integrating transcriptomics, proteomics and metabolomics data confirmed the flux re-routing predicted by the model. Taken together, our modelling and experimental results have implications for the computer-aided design of microbial cell factories.

**Keywords** adaptive laboratory evolution; chassis cell; metabolic engineering; multi-objective optimization; systems biology

**Subject Categories** Biotechnology & Synthetic Biology; Metabolism; Microbiology, Virology & Host Pathogen Interaction

**Mol Syst Biol. (2021) 17: e10253**

## Introduction

Design of an efficient yet robust microbial strain for producing molecules of interest is a major challenge in industrial biotechnology. Redirection of the nutrient influx to the target molecule requires multiple rounds of design and testing, and thereby substantial time and resources (Nielsen & Keasling, 2016). Advances in genetic engineering and systems biology have ushered in a spectrum of possibilities in the strain design process that goes beyond classical genome engineering, random mutagenesis and screening methods (Otero & Nielsen, 2010; Lee *et al*, 2011; Long *et al*, 2015). Yet, the identification of metabolic engineering strategies for re-routing intracellular fluxes towards a desired high production phenotype is not a straightforward task, mainly due to the complexity of metabolic networks. In the light of this challenge, the concept of chassis strains, i.e. microbial hosts pre-optimized for the production of a range of molecules, has been proposed towards reducing the cost of strain development (Vickers *et al*, 2010; Trinh *et al*, 2015; Jouhten *et al*, 2016). The concept of chassis builds upon the fact that despite the large chemical diversity of desired industrial compounds, most are derived from a limited set of precursor metabolites (Nielsen & Jewett, 2008). Consequently, modulation of native cellular metabolism to channel the carbon flux towards the required precursor and co-factors is key for the design of an optimal chassis strain (Vickers, 2016; Calero & Nikel, 2018).

Rational modulation of cellular metabolism requires accounting for the complexity of metabolic and regulatory networks. To this end, genome-scale metabolic models (GSMM), when combined with constraint-based algorithms (Savinell & Palsson, 1992; Segrè *et al*, 2002; Kauffman *et al*, 2003; Shlomi *et al*, 2005; Long *et al*, 2015), offer exciting possibilities for designing enhanced microbial strains (Patil *et al*, 2004; Adrio & Demain, 2006; Oberhardt *et al*, 2009; Agren *et al*, 2013). The advantages of strain engineering

1   Structural and Computational Biology Unit, European Molecular Biology Laboratory, Heidelberg, Germany
2   Life Science Institute, University of Michigan, Ann Arbor, USA
3   CEB-Centre of Biological Engineering, University of Minho, Campus de Gualtar, Braga, Portugal
4   Silicolife - Computational Biology Solutions for the Life Sciences, Braga, Portugal
5   Johann Wolfgang Goethe-Universität, Frankfurt am Main, Germany
6   The Medical Research Council Toxicology Unit, University of Cambridge, Cambridge, UK
7   Instituto de Tecnologia Química e Biológica António Xavier, Universidade Nova de Lisboa (ITQB-NOVA), Oeiras, Portugal
    *Corresponding author. Tel: +351 214469608; E-mail: irocha@itqb.unl.pt
    **Corresponding author. Tel: +44 1223 3 35640; E-mail: kp533@cam.ac.uk
    †These authors contributed equally to this work

strategies based on metabolic models include reduced trial-and-error, the possibility to integrate molecular and omics data in a structured manner, and the possibility to align cell growth with product formation. The latter is of particular interest since the product formation is often negatively correlated with cell fitness; consequently, productivity tends to drop over time due to fixation of adaptive mutations.

In *Saccharomyces cerevisiae*, a well-established microbial cell factory (Liu *et al*, 2013; Borodina & Nielsen, 2014; Li *et al*, 2015), model-guided strain design has been used for improving production of diverse compounds, including vanillin (Brochado *et al*, 2010), sesquiterpenes (Asadollahi *et al*, 2009) and dicarboxylic acids (Xu *et al*, 2012; Otero *et al*, 2013; Blazeck *et al*, 2014). Dicarboxylic acids are used as platform chemicals in the production of numerous high value-added compounds with widespread applications in the food, chemical and pharmaceutical industries (Werpy & Peterson, 2004; Patel *et al*, 2006; Roa Engel *et al*, 2008; Sauer *et al*, 2008). In this study, we report model-guided development of two yeast chassis strains for $C_4$-dicarboxylic acids, such as succinate, malate and fumarate. The producer cells derived from the chassis were challenged for their evolutionary stability and characterized at multiple omics levels to assess how closely the molecular changes aligned with the *in silico* predictions.

# Results

### Metabolic modelling for $C_4$-dicarboxylic acid chassis

We used a genome-scale metabolic model-based strategy to identify genetic targets for chassis design. Yeast metabolic model iMM904 (Mo *et al*, 2009) was used to represent the biochemical capabilities and inter-connectivity of the yeast metabolism. The iMM904 model was updated according to the yeast consensus model (Yeast 7.11) and to include known mitochondrial transporters. The resulting modified iMM904 model consists of 1,417 reactions, 1,064 internal metabolites and 904 genes (Materials and Methods, Appendix Table S1). We then used a multi-objective metaheuristic approach to search for gene knockouts that would couple cell growth to product formation. In brief, the algorithm searches for network modifications (in this case, one or more gene knockouts) such that the optimal flux distribution for biomass formation generates overflow of the target molecule (Burgard *et al*, 2003; Patil *et al*, 2005; Rocha *et al*, 2008) (Materials and Methods and Fig EV1). These solutions were then clustered to identify frequently occurring sets of gene targets common to all three products (Fig 1A). The solutions with > 95% of maximum carbon yield for each target compound were ranked by the ratio between predicted carbon yield and the required number of gene knockout (Chassis score) (Appendix Tables S2–S4). This raking thus prioritizes solutions with higher carbon yield per network modification (Materials and Methods). The developed framework was applied to search for solutions to engineer a yeast chassis strain towards enhanced biosynthesis of three target $C_4$-dicarboxylic acids: succinic acid, fumaric acid and malic acid. The robustness of the predictions was verified across different versions of yeast genome-scale metabolic models (iMM904, iND750 and Yeast6) and against different simulation methods (pFBA and lMOMA) (Appendix Tables S5–S6).

A common observation in the identified *in silico* solutions was that *ZWF1* deletion had a significant contribution to the biomass-coupled production of all three target products. The top-scoring chassis design solution suggested deletion of three genes, *ZWF1*, *SER3* and *SER33*, with predicted product yields of nearly half of the maximum theoretical values and biomass yield only reduced to about one third of the reference (Appendix Fig S1). *ZWF1* encodes glucose-6-phosphate dehydrogenase, the first enzyme in the pentose phosphate pathway (PPP) (Nogae & Johnston, 1990), while *SER3* and *SER33* (3-phosphoglycerate dehydrogenase isozymes) catalyse the first reaction in the serine biosynthetic pathway starting from the glycolytic intermediate 3-phosphoglycerate (Fig 1A and B) (Albers *et al*, 2003). In a *Δser3,33* mutant, serine, an essential amino acid, needs to be synthesized from glycine through the serine hydroxymethyltransferase reaction. The required glycine needs then to be produced either from the high NADPH demanding threonine biosynthetic pathway, via the low-specificity threonine aldolase (Gly1p,) or from glyoxylate by alanine:glyoxylate aminotransferase (Agx1p). Increased flux through the glyoxylate shunt—which converts acetyl-CoA into succinate, glyoxylate and malate—can thus couple serine biosynthesis to succinate and malate production. Indeed, the *Δser3,33* mutant was previously shown to couple succinate production to growth when combined with the deletion of *SDH3* (Otero *et al*, 2013). However, it was not previously suggested or tested whether the same strategy would work for the production of fumarate and malate.

The next model-predicted solution, the inactivation of the oxidative pentose phosphate pathway through *ZWF1* deletion, would disrupt the main source of cytosolic NADPH. This is predicted to result in an increased flux through other cytosolic NADPH generating reactions, such as isocitrate dehydrogenase, aldehyde dehydrogenase and succinate semialdehyde dehydrogenase (last enzyme in GABA catabolism, generating both NADPH and succinate). Since a significant growth impact is expected in a *ZWF1* knockout strain (Partow *et al*, 2017), we implement two chassis designs: (i) deletion of *SER3,33* ("Chassis"); and (ii) deletion of *SER3,33* combined with that of *ZWF1* ("Chassis_z"). The latter strategy would simultaneously interrupt the oxidative branch of the pentose phosphate pathway and the serine/glycine production from glycolysis to increase flux to TCA cycle and glyoxylate shunt.

To convert the chassis strains into producer strains, additional knockout targets were predicted for each product: succinate dehydrogenase (*SDH-complex: SDH1, SDH2, SDH3 or SDH4*) for succinate, fumarate hydratase (*FUM1*) for fumarate and malate dehydrogenase (*MDH1* and *MDH2*) and *MAE1* (which catalyses the oxidative decarboxylation of malate to pyruvate) for malate (Fig 1A and B, Appendix Fig S1B–G and Appendix Table S2–S4). All these product-specific solutions will disrupt the conversion of the target molecule in the TCA cycle and thereby predicted to result in the secretion the target compound.

### Chassis strains do not accumulate dicarboxylic acids

Deletions of the identified targets for the two chassis designs—*SER3*, *SER33* and *ZWF1*—were engineered in a *S. cerevisiae* CEN.PK background strain (Fig 1C, Materials and Methods). The resulting "Chassis" (*Δser3,33*) and "Chassis_z" (*Δser3,33Δzwf1*) strains were characterized for growth and target compound production in

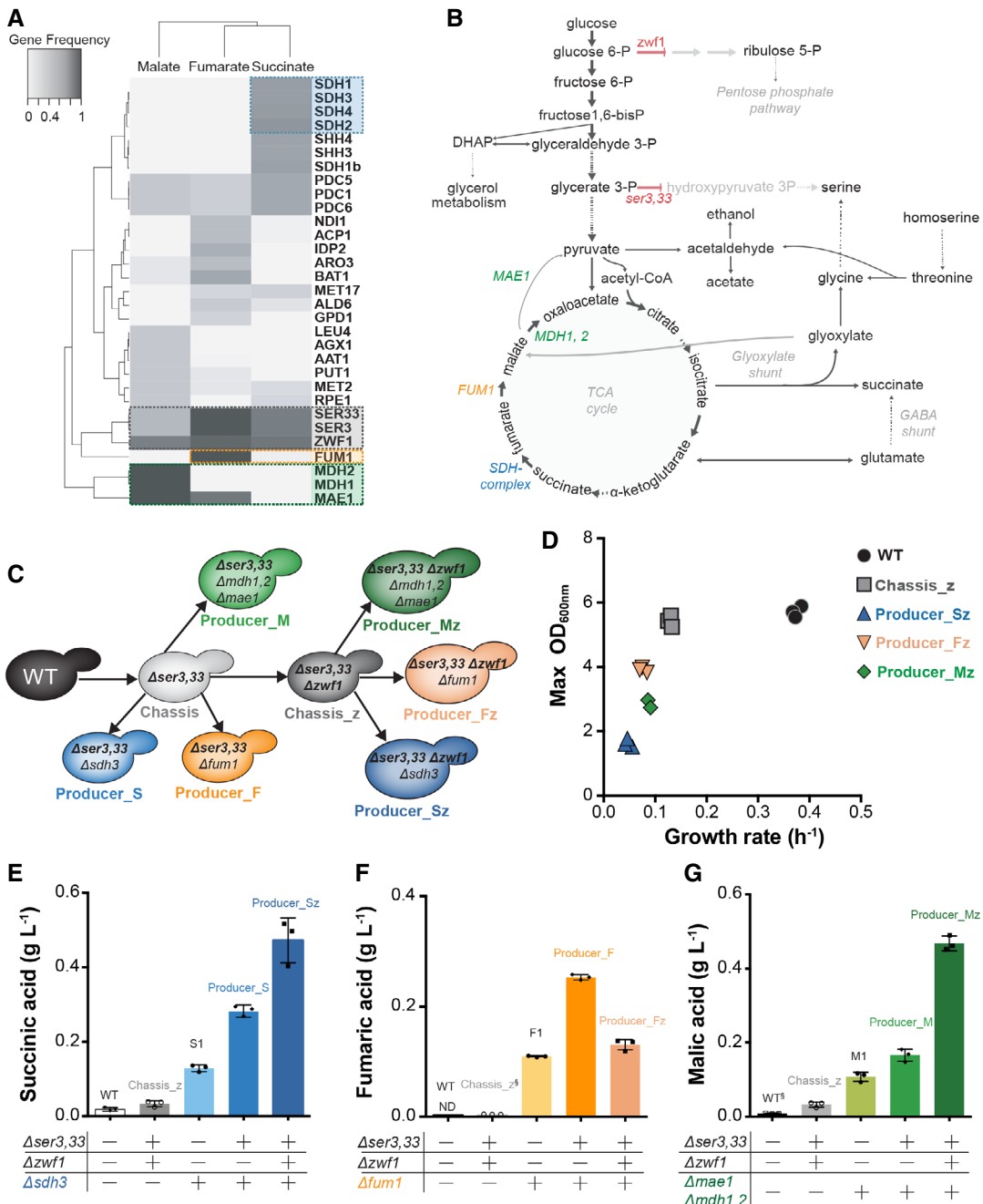

**Figure 1. Gene frequency analysis of chassis-strain design, engineering of knockout solutions *in vivo* and compound production phenotype.**

A   Hierarchical clustering of gene clusters (rows) computed over the gene frequencies for each of the target products (column)—malic acid, fumaric acid and succinic acid. Frequencies below 20% were excluded. The selected Chassis_z backbone is represented in grey (Δ*ser3*, Δ*ser33* and Δ*zwf1*). The blue, green and orange clusters denote suggested gene deletions to obtain producer strains from the chassis, for succinate, malate and fumarate, respectively.

B   Metabolic pathways involved in target compound production and predicted carbon fluxes in Chassis_z. Gene targets predicted by the model for chassis design are coloured in red. Red arrows represent deleted reactions, and arrow thickness represents relative flux values for Chassis_z (simulated with pFBA). Green, orange and blue gene names show the knockout reactions to obtain the producing strains for malate, fumarate and succinate, respectively. Dashed arrows represent multiple reactions.

C   Scheme of strains engineered in this study.

D   Growth rate and maximum OD obtained for engineered chassis_z-derived Producer, Chassis_z and wild-type strains (Each symbol represents an independent biological replicate, $n = 3$).

E–G   Production of the target organic acid compound—succinic acid (E), fumaric acid (F) and malic acid (G)—by wild-type, TCA disrupted (S1: Δ*sdh3*, F1: Δ*fum1* and M1: Δ*mae1*,Δ*mdh1,2*), Chassis_z, chassis- and chassis_z-derived producing strains. Each symbol represents an independent biological replicate—$n = 3$, bars are average values and error bars denote standard deviation. ND—metabolite not detected; and §—metabolite identified but not quantified due to low AUC. All cultures were performed in minimal media with 2% glucose, and chassis_z-derived Producer strains were supplemented with glycine.

Source data are available online for this figure.

    

minimal media with 2% glucose (Figs 1D and EV2A and E–G). While the growth rate of the Chassis_z strain decreased compared with the wild-type and Chassis strains, the biomass yield of these strains remained similar (Figs 1D and EV2A). As predicted by the model, both Chassis and Chassis_z strains did not accumulate any appreciable amounts of extracellular $C_4$-dicarboxylic acids (succinic, malic and fumaric acid) (Figs 1E–G and EV2E–G).

## Strains built from the chassis secrete dicarboxylic acids in concordance with model predictions

As per predictions from flux modelling, product-specific gene deletions would be required to generate producer strains from the chassis. We engineered the corresponding gene deletions ($\Delta sdh3$, $\Delta fum11$ or $\Delta mae1$, $\Delta mdh1,2$) in both chassis strains to test for the production of succinic, fumaric and malic acids, respectively (Fig 1 C, Appendix Table S7). All engineered strains were characterized for growth (Figs 1D and EV2A–D) and metabolite production in minimal media (Figs 1E–G and EV2E–G, Appendix Tables S10 and S11). As secretion of target products can be expected by mere TCA cycle disruption, the product-specific gene deletions were also engineered in a wild-type yeast strain. These control strains, S1 ($\Delta sdh3$), M1 ($\Delta mdh1,2$ $\Delta mae1$) and F1 ($\Delta fum1$) are able to secrete 0.13 g/l succinic, 0.11 g/l malic and 0.11 g/l fumaric acid, respectively (Figs 1E–G and EV3E–G). As predicted, the disruption of the TCA cycle in the chassis background led to a substantial further increase in the secretion of the three target compounds. The highest succinic acid titre (0.47 g/l) was observed for the Producer_Sz strain ($\Delta ser3,33\Delta sdh3\Delta zwf1$), a threefold increase over the S1 strain and twofold over the strain without ZWF1 deletion (Producer_S: $\Delta ser3$, ser33 $\Delta sdh3$) (Figs 1E and EV2E). Similarly, the malic acid producer strain, Producer_Mz ($\Delta ser3,33$ $\Delta zwf1$ $\Delta mdh1,2$ $\Delta mae1$), showed a 2.5-fold increase in malic acid titre (0.47 g/l) compared with Producer_M ($\Delta ser3,33$ $\Delta mdh1,2$ $\Delta mae1$) and a 4.7-fold increase compared with the control strain M1 (Figs 1G and EV2G). Deletion of FUM1 in Chassis and Chassis_z background led to a 2.3- and 1.2-fold increased secretion of fumaric acid when compared to F1 control strain, respectively. Despite *in silico* predictions, the deletion of FUM1 in Chassis_z background (Producer_Fz: $\Delta ser3,33$ $\Delta zwf1$ $\Delta fum1$) resulted in 1.5-fold less titre than in the Producer_F strain ($\Delta ser3,33$ $\Delta fum1$) (0.25 g/l) (Figs 1F and EV2F). Nevertheless, the contribution of both chassis designs is evident for the enhanced production of the target molecules.

The best producing strains—Producer_Sz, Producer_Mz and Producer_F—were characterized at transcriptomics and proteomics level to assess whether the changes at molecular level are in accordance with model predictions. Wild-type strains cultivated under the same conditions were used as control. A key model prediction was increased flux through the glyoxylate shunt due to disruption of serine biosynthesis via glycolysis. Glyoxylate is produced from isocitrate by isocitrate lyase (encoded by ICL1). Next, glyoxylate is converted to glycine by alanine:glyoxylate aminotransferase (encoded by AGX1). All three engineered producer strains showed upregulation of AGX1 at the transcriptome and proteome level. On the other hand, ICL1 expression levels were not found significantly altered in the engineered producer strains. In the presence of glucose, deletion of the regulatory subunit of the protein phosphatase type 1 (REG1) has a higher impact on Icl1p enzymatic

activity than ICL1 overexpression (Koivistoinen *et al*, 2013). Interestingly, overexpression of ICL1 was previously described to enhance succinate production, but only in strains previously evolved to improve flux trough glyoxylate shunt (Otero *et al*, 2013). Producer_Sz and Producer_Mz strains had significantly lower Reg1 protein abundances than the wild-type strain (1.8 and 1.7 log2fold change, respectively, with adjusted $P < 0.05$). Producer_F also present a lower Reg1 protein abundance albeit less significantly (0.9 log2fold change, adjusted $P < 0.05$). In engineered strains, the observed downregulation of Reg1p can positively affect Icl1p activity. These results suggest that the predicted increased isocitrate-to-glycine flux was attained at both enzymatic steps (Icl1p and Agx1p). The second key model prediction is the increased flux though the GABA shunt for NADPH production in ZWF1 deleted strains. In accord, the chassis-derived producer strains deleted in ZWF1 (Producer_Sz and Producer_Mz) have Uga1p and Uga2p (second and third enzymatic steps in GABA shunt) up-regulated, while in the engineered strain not deleted in ZWF1, Producer_F, the protein amounts of these enzymes are not changed. These results show concordance between *in silico* model predictions and altered transcript and enzyme abundances in glyoxylate and GABA shunts.

While the enhanced production in concordance with model predictions was encouraging, the combination of TCA cycle and serine biosynthesis disruption caused the strains to be auxotrophic for serine. While serine auxotrophy can be alleviated by glycine supplementation, it is not desirable from application perspective. Further, all chassis-derived producing strains exhibited lower growth rates than the chassis, the wild-type and control strains (Figs 1D and EV2A–D). The serine auxotrophy and reduced growth rates show that, despite increased production and concordant protein abundance changes, significant divergence between the model predictions and in vivo changes remained.

## Adaptive laboratory evolution restored glycine prototrophy and improved fumaric and succinic acid production

The model predictions were based on assumption of optimal flux re-routing in the engineered strains. However, re-organization of the metabolic fluxes to compensate for the effect of gene knockouts may require evolutionary adaptation (Szamecz *et al*, 2014). Therefore, to overcome the growth deficiencies of the engineered strains, we resorted to another aspect of the model predictions, namely the coupling between growth and production. We hypothesized that adaptive evolution would lead to mutants with not only improved fitness, but also production. Additionally, adaptive evolution would allow us to assess the stability of compound production in producer strains. To test this, we subjected the engineered Chassis_z and Chassis_z-derived producer strains to adaptive laboratory evolution. From the Chassis-derived strains, Producer_F ($\Delta ser3,33$ $\Delta fum1$) was also included in the adaptive laboratory evolution experiment, since this strain had the best fumarate producing phenotype among all our engineered strains. Cells from three single colonies of Chassis_z, Producer_Sz, Producer_Mz, Producer_Fz and Producer_F were evolved in parallel in minimal media, only supplemented with glycine in the first 4 serial passages (Appendix Fig S2A). At each passage, optical density (OD 600nm) and transfer time were recorded (Appendix Fig S2B–E). All three independent populations of Producer_Fz lost the ability to produce fumaric acid (< 0.01 g/l)

after only 8 passages. After 24 serial passages, adapted populations were grown on minimal media agar plates and 3 isolated colonies were evaluated for fitness and target compound production (Appendix Fig S3A–C). The isolate showing the highest target compound production, per parental strains, was further characterized for growth and metabolite production (Figs 2A–D and EV3A–D, Appendix Table S11). The evolved strains not only exhibited improvement in the specific growth rate, but also, in accord with the model predictions, could grow without glycine supplementation (Fig EV3A–D). The evolved succinate producer strain (E_Producer_Sz) showed a notable sixfold improvement in the specific growth rate (Appendix Fig S3B and Appendix Table S10). The evolved fumarate and malate-producing strains, E_Producer_F and E_Producer_Mz, showed a 4.6-fold and 1.4-fold improvement, respectively (Appendix Fig S3C and D, Appendix Table S10). Adaptive laboratory evolution also resulted in a substantially improved production of succinate and fumarate (circa 1.5-fold), but not in malate (Fig 2A–C, Appendix Table S10). Moreover, the improvement in succinate titre (> 1.1 g/l) was maintained in the evolved strains when tested in batch reactors under controlled conditions (Appendix Table S10).

Production of non-target dicarboxylic acids (i.e. fumarate and malate) was not observed in the succinate producer strains before nor after evolution (Appendix Table S11). On the other hand, fumarate producer strain (Producer_F) also secreted succinate, albeit in much less amounts than the succinate producing Producer_Sz strain. Nevertheless, after evolution, fumarate production in Producer_F increased by 70%, while the succinate production decreased by 60%. The robustness of growth-coupled malate production was only predicted when fumarate drain was impaired (Appendix Fig S1G). *In vivo*, malate producer strain secreted fumarate in low amounts (~0.04 g/l), which further dropped after evolution (Appendix Table S11). Overall, these results support the evolutionary robustness of the growth-coupled strain design.

**Metabolic changes after evolution approach *in silico* predictions**

The improved growth rates and changes in compound production after evolution (Fig 2D) prompted us to explore how the evolved strains responded to the growth-production dependency. We characterized the evolved producer strains and the corresponding parental strains using multi-omics analyses (Fig EV4).

Whole genome sequencing of the evolved strains revealed only a small number of single nucleotide (SNV) and insertion-deletion (indel) variants in coding sequences (CDS); structural variations or aneuploidy were not detected (Appendix Fig 7B). We did not find any SNVs and indel that recurred in all evolved strains. The evolved succinate producer strain, E_Producer_Sz, harboured the highest number of mutations (6 in total). Three of the six mutations concern regulatory proteins, including *GCN5* (Subunit of SAGA complex) and *UBC8* that are known players in glucose utilization and redox balance. Two SNVs were identified in metabolic genes, *ERG5* and *GLY1*. A SNV in *GLY1* was also identified in the evolved malate producer strain E_Producer_Mz. Three out of four mutations identified in this strain are in regulatory proteins. One of these is a frameshift mutation in *CTI6*—a regulatory protein required to relieve transcriptional repression of glucose-repressed genes. The evolved fumarate producer strain, E_Producer_F, has also 4 SNVs (*TRS85*,

*STE20*, *RTK1* and *TOR1*), with no targets shared with the other evolved producer strains. However, a frameshift mutation in the putative kinase *RTK1* was also identified in the evolved chassis strain E_Chassis_z. In general, the observed number of mutations, as well as the proportion of metabolic/regulatory targets are similar across the different evolved strains, hinting that the re-routing of metabolic fluxes towards optimality was mainly driven by changes at regulatory level.

At the gene expression level, the numbers of significantly ($1 \geq$ or $-1 \leq \log_2$fold change with adjusted $P < 0.05$) up- and down-regulated genes in the evolved strains were similar (succinate: 86/263, fumarate: 225/210, malate 280/194). Principle component analysis (PCA) showed that all parental strains cluster together with some separation along PC2 driven by *ZWF1* deletion (Fig EV4C). After evolution, succinate and fumarate producer strains cluster closer together, separated from their parental strains. In contrast, the evolved malate-producing strain moved closer to the wild-type (Fig EV4C).

The number of proteins with significantly ($1 \geq$ or $-1 \leq \log_2$fold change with adjusted $P < 0.05$) changed abundances are smaller than the changes at the gene expression level (succinate: 389, fumarate: 186, malate: 186). As in the case of gene expression, parental succinate and malate producer strains, both deleted in *ZWF1*, were more similar to each other than to the parental fumarate strain (Fig EV4D). After evolution, succinate and fumarate producer strains cluster closer together, again reflecting the trend observed in the transcription data (Fig EV4C). In the case of the malate-producing strain, the evolved and parental strains were very similar (Fig EV4D). Gene-set enrichment analysis showed that most changes in the evolved strains, at gene expression as well as protein abundance levels, were related with amino acid, energy, glycolysis and carboxylic acid metabolism (Appendix Fig S4). Overall, the small number of identified genetic mutations and relatively high number of differentially regulated transcript and protein abundances suggest that the phenotypic changes following evolution are driven by changes in the regulatory network. Targeted metabolomics analysis (Fig EV4E) showed that evolution did not dramatically change the exometabolome, except for the target product, and the evolved strains cluster with the respective parental, with the exception of the E_Producer_Mz. In this strain, the exometabolome after evolution is closer to wild-type than to its parental strain (Producer_Mz), in accord with the transcriptomics and the decreased malic acid secretion.

Next, we integrated the multi-omics data with flux balance analysis to refine flux phenotype prediction. For this, we used relative changes in transcript and protein abundances between two conditions as constraints on the corresponding reaction rate (Machado *et al*, 2016). In addition, changes in extracellular metabolite concentrations were used to constrain the uptake/secretion of the corresponding metabolites. A flux balance solution consistent with these constraints was obtained using linear programming. The flux phenotype of the parental strain was predicted by comparing the omics data of the engineered producer strains to those of the wild-type, whereas the evolved strains were compared with their respective parental strains (Materials and methods). PCA of the predicted flux distributions shows that the multi-omics data are consistent with model predictions (Fig 2E). Moreover, after evolution all producer strains are closer to the model predictions than the parental

producer strains (Fig 2E). The metabolic changes driven by evolution are thus in line with the *in silico* predictions, validating the theory underlying the chassis-strain design.

## Omics data uncover model shortcomings in fumarate and malic acid metabolism

Multi-omics data were also used to find the metabolic routes taken by the best performing strains E_Producer_F and E_Producer_Sz, and to identify the mechanisms underlying the decreased production observed in the evolved E_Producer_Mz and E_Producer_Fz strains (Fig 3). In the evolved succinate producer strain, increased transcription of *GAD1* and *UGA2* (encoding glutamate decarboxylase and succinate semialdehyde dehydrogenase, respectively) suggested an increased α-ketoglutarate-to-succinate conversion through the GABA shunt yielding more NADPH. Increased abundances of *GAD1* and *UGA2* was also observed in the strain E_Producer_F at both transcript and proteome level, as well as increased levels of GABA transporters (*UGA4*, *PUT4* and *GAP1*). Thus, even though the oxidative phosphate pentose pathway, the main source of NADPH

production in yeast, was not disrupted, the flux through the GABA shunt was increased to enhance NADPH availability.

Two proteins were up-regulated in all 3 evolved producer strains, namely Arg4p and Gly1p (Fig 3). Argininosuccinate lyase (Arg4p) catalyses the conversion of L-arginino-succinate to L-arginine and fumarate, while threonine aldolase (Gly1p) converts L-threonine to glycine and acetaldehyde. Characterization of the parental producer strains was performed in media supplemented with glycine, which is in contrast to the evolved and wild-type strains that are not auxotrophic for glycine. Yet, increased abundance of GLY1p was not observed in wild-type cells suggesting that the increase in Gly1p in the evolved producer strains is not associated with growth conditions, but with its use for glycine biosynthesis. In E_Producer_Sz and E_Producer_Mz, the increased protein abundance is coupled with a SNV in the *GLY1* coding sequence. The E_Producer_Mz strain also had *AGX1* down-regulated at the protein level, suggesting a reduction in glycine biosynthesis via glyoxylate shunt. Increased abundances of Mdh3p, in both parental and evolved malate-producing strains compared with WT, suggest that in the absence of Mdh1p and Mdh2p, this enzyme is able to carry flux (Steffan & McAlister-Henn, 1992). We conclude that

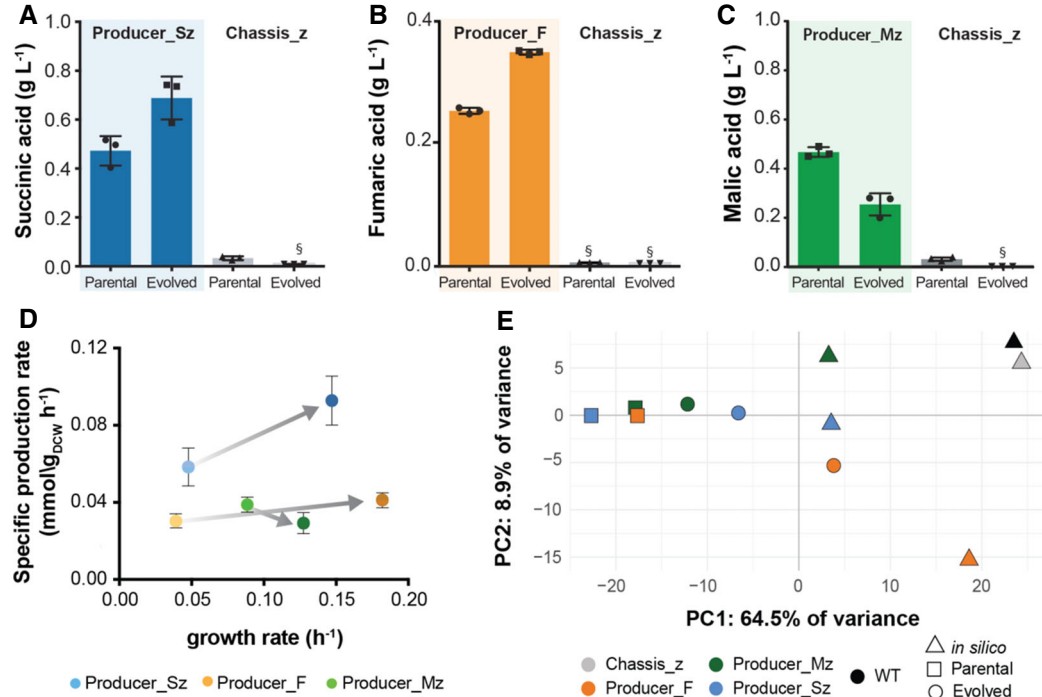

**Figure 2.  Target compound production in evolved and parental producer strains, and multi- metabolic flux predictions based on omics data.**

A–C   Production of respective target compounds by Chassis_z strains parental and evolved. (A) Succinic acid production by parental (Producer_Sz) and evolved (E_Producer_Sz) succinate producer strain. (B) Fumaric acid production by parental (Producer_F) and evolved (E_Producer_F) fumarate producer strain. (C) Malic acid production by parental (Producer_Mz) and evolved (E_Producer_Mz) producer malate strain.

D   Specific target compound production and growth rates of evolved and parental succinate (blue), fumarate (orange) and malate (green) producing strains. Grey arrows connect parental to the evolved strains. Average of independent replicates are shown, $n = 3$. Error bars denote standard deviation.

E   Principle component analysis of metabolic flux simulations, integrating multi-omics data, for parental (squares) and evolved (circles) producer strains. Triangles show *in silico* model predictions on which the producer strains were based.

Data information: (A–C) Each symbol represents an independent biological replicate—$n = 3$, bars are average values and error bars denote standard deviation. All cultivations were performed in minimal media with 2% glucose, and parental strains were supplemented with glycine. §—metabolite identified but not quantified due to low AUC.

Source data are available online for this figure.

the improved glycine production from L-threonine, together with decreased Agx1p and increased Mdh3p abundances, allowed the Producer_Mz strain to circumvent the growth-coupling of malate production during directed evolution (Fig 3).

Gdh2p (NADH-dependent glutamate dehydrogenase) was down-regulated in both E_Producer_Mz and E_Producer_Sz, with the later strain also featuring Gdh1p (NADPH-dependent glutamate dehydrogenase) upregulation. Divergently, Gdh2p was up-regulated in the

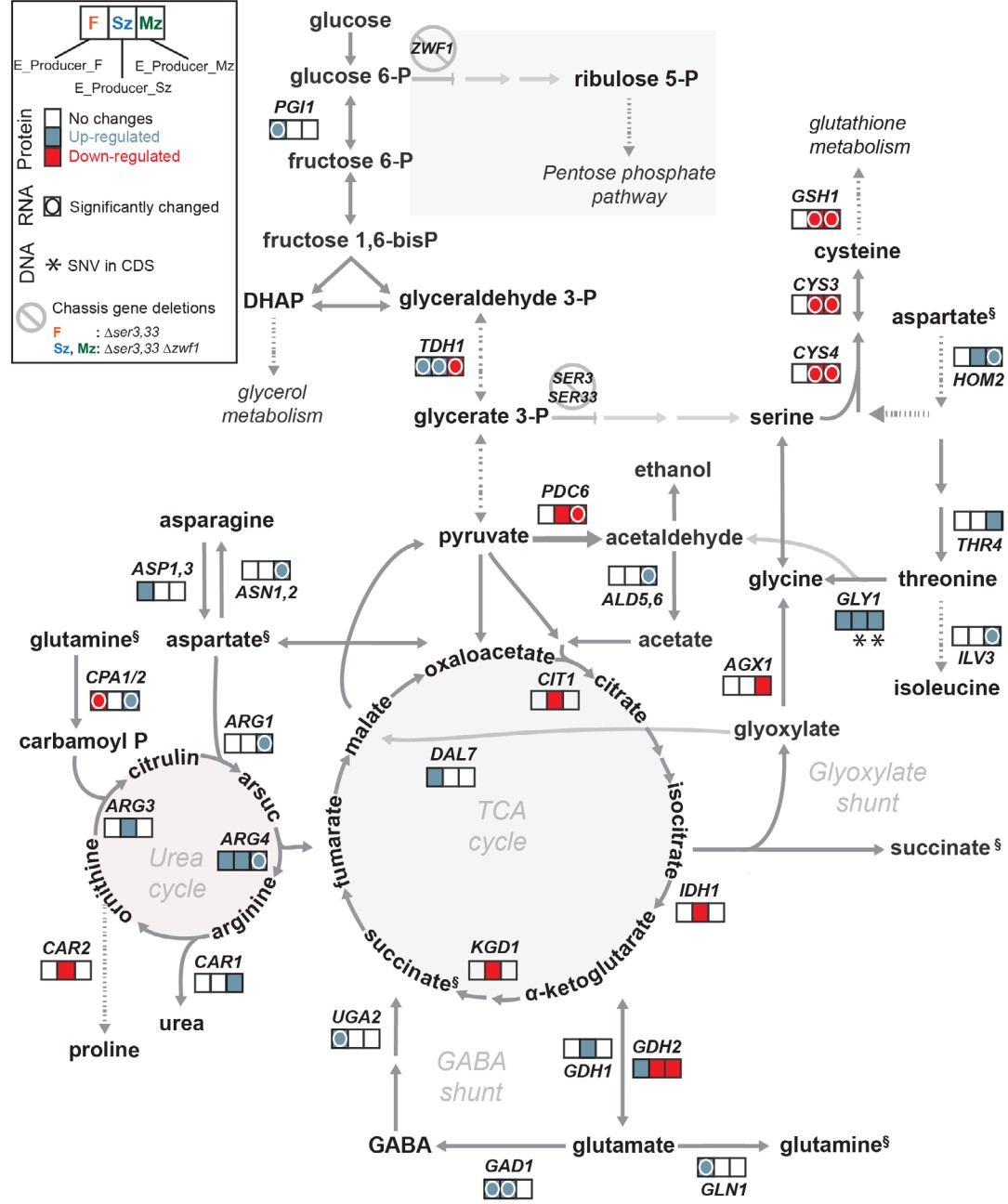

**Figure 3. Metabolic network showing genomics, transcription and proteomics analysis of relevant pathways in chassis-derived organic acid producer strains after evolution.**

Proteomics, transcriptomics and genomics data are shown next to the corresponding reaction as abundance ratios between the evolved and the parental strains (The three squares next to reaction represent fumaric, succinic and malic acid producer strains—left to right). Only enzymes with protein changes observed in at least one of the three strains are shown. Significant changes ($1 \geq$ or $-1 \leq \log_2$fold change with adjusted $P$-value (moderated $t$-test with Benjamini–Hochberg false discovery rate adjustment) < 0.05) in protein abundances are depicted by coloured squares. Genes with identified SNVs in CDS are marked with (*). Gene expression changes are marked with (O) if transcriptomics data are in concordance with proteomics data. §—Metabolites represented twice. Dashed arrows represent series of enzymatic steps and full arrows one enzymatic step. Abbreviations: DHAP—dihydroxyacetone phosphate, GABA—γ-aminobutyric acid.

Source data are available online for this figure.

strain E_Producer_F (Fig 3). Engineering of *GDH1* deletion and *GDH2* overexpression has been shown to rescue phenotypes associated with low NADPH availability (Asadollahi *et al*, 2009; Brochado *et al*, 2010). We therefore investigated whether the fumarate producing strain deleted in *ZWF1*, Producer_Fz, decreased fumaric acid production during evolution due to low NADPH availability (Fig 4A).

### Engineering glutamate dehydrogenase node improved fumarate production and evolutionary stability

The glutamate dehydrogenase node was engineered in Producer_Fz (Δ*ser3,33* Δ*zwf1* Δ*fum1*) by deleting *GDH1* and overexpressing *GDH2*. Supporting our prediction, the resulting strain, Producer_FzG (Δ*ser3,33* Δ*zwf1* Δ*fum1* Δ*gdh1::GDH2*), produced 0.165 g/l of fumaric acid, an improvement of 1.3-fold. Importantly, this increase in production was not offset by growth rate reduction; moreover, the lag-phase was considerably shortened (Fig 4B and C). To test evolutionary stability, both Producer_Fz and the Producer_FzG were subject to adaptive laboratory evolution (Materials and Methods,

Appendix Fig 2A). After 24 passages, isolates from the evolved populations were characterized in minimal media without glycine supplementation. While a striking loss of fumarate production was observed in the case of Producer_Fz, the evolved isolates of E_Producer_FzG showed ~1.5-fold improvement in fumarate production over their parental strain (Fig 4D). These results, in combination with the observation that the fumarate strain not deleted in ZWF1 (i.e. Δ*ser3,33* Δ*fum1*) successfully adapted for improved fumarate production and fitness (Fig 2C), suggest that fumaric acid production requires higher NADPH supply than predicted *in silico*. Re-engineering of the glutamate dehydrogenase node (Δ*gdh1::GDH2*) was able to buffer the redox changes during evolution, leading to an improved growth and production phenotype (Fig 4C and D).

We observed a very high number of mutations (> 300 SNV and Indels), distributed throughout the genome, in E_Producer_Fz strain. The number of mutations observed in this strain is around 100 times higher than identified in any other evolved strains and consistent with the loss of fumarate production (uncoupling growth and production). The deletion of *FUM1* has been associated with

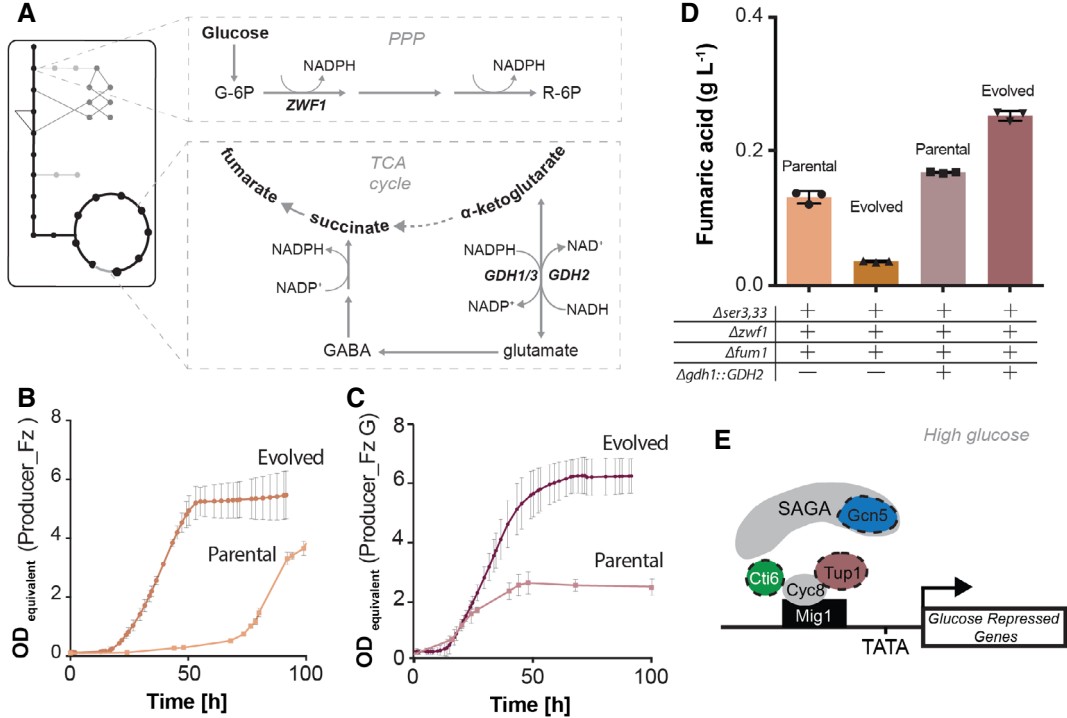

**Figure 4. Impact of glutamate node engineering in fumarate producer strain.**

A Glutamate node, identified by multi-omics strain characterization as target for engineering, is a major hub of NADPH metabolism (Abbreviations: G6P—glucose 6-phopsphate, R6P—ribulose 6-phosphate, GABA—γ-aminobutyric acid, PPP—pentose phosphate pathway).

B Growth phenotype of parental (Producer_Fz) and evolved (E_Producer_Fz) fumarate producer strains. Parental strain was supplemented with glycine (*n* = 3 biological replicates, average values are presented and error bars represent standard deviations).

C Growth phenotype of parental (Producer_FzG) and evolved (E_Producer_FzG) fumarate producer strains engineered in glutamate node (Δ*gdh1::GDH2*). Parental strain was supplemented with glycine (*n* = 3 biological replicates, average values are presented and error bars represent standard deviations).

D Fumaric acid production by parental (Producer_Fz) and evolved (E_Producer_Fz) producer strain compared with parental (Producer_FzG) and evolved (E_Producer_FzG) producer strain engineered in glutamate node. Each symbol represents an independent biological replicate—*n* = 3, Bars are average values, and error bars denote standard deviation. All cultures were performed in minimal media with 2% glucose, and parental strains were supplemented with glycine.

E Adapted model of Tup1/Cyc8-Cti6-Gcn5 interaction for altered transcription of glucose-repressed genes (Papamichos-Chronakis *et al*, 2002). Dashed proteins represent indel mutations identified in evolved strains, Gcn5p (blue—E_Producer_Sz), Cit6p (green—E_Producer_Mz) and Tup1p (pink—E_Producer_FzG).

Source data are available online for this figure.

increased sensitivity to DNA damage which can manifest into hyper-mutation phenotype (Yogev *et al*, 2010). In our evolution experiments, the hyper-mutation phenotype (300 SNVs) is apparent only in one of the three Δ*fum1 strains* (E_Producer_Fz: Δ*ser3,33* Δ*zwf1* Δ*fum1*). In the remaining Δ*fum1* strains, the one lacking *ZWF1* deletion (E_Producer_F) and the one engineered for improved NADPH availability (Producer_FzG), the number of identified mutations is comparable to the low number of mutations in the other evolved producer strains (< 5 SNVs) (Fig EV4B). The hyper-mutation phenotype in the E_Producer_Fz yeast strain thus appears to be contingent on redox imbalance. In E_Producer_FzG strain (Fig EV4B), a SNV was identified in the coding region of *HOM3*, the second enzymatic step in the aspartate-to-threonine pathway, and two frameshift mutations were identified in *NMD2* and *TUP1*. Tup1p is a general transcriptional repressor, which forms a complex with Cyc8p involved in catabolite repression. In high glucose conditions, the complex Tup1-Cyc8 associates with transcriptional regulators repressing the expression of target genes. The interaction between the Tup1-Cyc8 complex with Cti6 and the SAGA complex is responsible for de-repression under low glucose conditions (Papamichos-Chronakis *et al*, 2002). Interestingly, indel mutations were observed in *CIT6* and *GCN5* (catalytic subunit of SAGA complex) of E_Producer_Mz and E_Producer_Sz, respectively (Fig 4E, Appendix Fig S4B). These suggest that a modified assembly of this complex—driven by the identified indel mutations—allowed a fine-tuned transcription of glucose-repressed genes towards an optimal metabolic flux distribution during evolution.

# Discussion

Model-guided design of chassis microbial strains has the potential to overcome the cost and time burden associated with cell factory development, but so far has only been explored theoretically (Layton & Trinh, 2014; Trinh *et al*, 2015; Jouhten *et al*, 2016). In this work, a rational strategy based on first-principle modelling was used to develop two *S. cerevisiae* chassis strains for the enhanced production of dicarboxylic acids. The contribution of the chassis background to the production of the three dicarboxylic acids is evident for all the target products. Disruption of single reaction steps in the TCA cycle in the chassis background increased fumarate, succinate and malate titre—2.3, 3.7 and 4.4-fold—in comparison with the respective disruption of the TCA cycle in a wild-type background. The chassis-strain design, presented here, demonstrates that genome-scale metabolic models can identify non-intuitive metabolic routes for building pre-optimized chassis and subsequent improvement in product flux. Our results set a stage for further exploration of modelling capabilities through model designs accounting for regulatory networks and other cellular processes (Sánchez *et al*, 2017; Lloyd *et al*, 2018).

Engineered microbial cells require evolutionary adaptation to the implemented network changes. While adaptive evolution can be used to improve phenotype stability, robustness and fitness for biotechnological applications (Fong & Palsson, 2004; Strucko *et al*, 2018; Yu *et al*, 2018), it can also lead to loss of production due to natural selection for better fitness. Modelling strategies based on growth-product coupling can mitigate this risk as demonstrated in our work. Indeed, adaptive laboratory evolution of chassis-derived producing strains not only alleviated auxotrophy and improved

growth fitness, but also led to a 1.4-fold increase in succinate and fumarate production. In the opposite direction, malate production decreased after evolution.

Multi-omics strain characterization uncovered a synergetic effect of Gly1p, Agx1p and Mdh3p changed abundances as a possible route for the circumvention of the growth-coupling of malate production. The model simulations used for the design of the chassis-derived malate-producing strain did not detect flux for malate-to-oxaloacetate conversion by the peroxisomal malate dehydrogenase (Mdh3p). Yet, this reaction is present in the model and *in vivo* accounts for a fraction of the total cellular malate dehydrogenase activity (Steffan & McAlister-Henn, 1992). We investigated what limits, *in silico*, the malate-to-oxaloacetate flux in the peroxisome and identified that both yeast models—iMM904 and consensus yeast 7.11—lack the peroxisomal glycerol-3-phosphate dehydrogenase (encoded by *GDP1*) reaction, required to balance NADH/NAD$^+$ in peroxisomes (Al-Saryi *et al*, 2017). Since the NADH imbalance restricts malate dehydrogenase activity in the peroxisome, *MDH3* was not detected in model simulations as a target deletion solution for improved malate production. Based on these findings, peroxisomal metabolism was updated in the yeast metabolic model—iMM904—to include glycerol-3-phosphate dehydrogenase reaction (Materials and Methods). With this update, flux balance analysis of the Producer_Mz strain genotype (Δ*ser3,33*Δ*zwf1*Δ*mae1*Δ*mdh1,2*) predicted flux through the reactions encoded by *MDH3* and GDP1 and no malate secretion, as expected. Malate secretion phenotype was predicted when all three malate dehydrogenase isoenzymes (Mdh1,2,3) were deleted.

Fumarate titre was 1.9-fold higher in Producer_F (Chassis background, Δ*ser3,33* Δ*fum1*) than in Producer_Fz (Chassis_z background, Δ*ser3,33* Δ*zwf1* Δ*fum1*), conversely to model predictions. Moreover, after evolution Producer_Fz lost fumarate production. Engineering the glutamate node, indicated by the multi-omics analysis of the evolved strains, allowed us to improve production and evolutionary stability of the fumarate producer strain. The deviations from the model predictions in both the fumarate and malate cases identify redox metabolism as one of the limitations that will need to be addressed for improving model-guided strain design.

Whole genome sequencing of the evolved producer strains deleted in *ZWF1* revealed mutations in proteins linked to glucose repression of metabolic genes (*TUP1*, *CTI6* and *GCN5*) (Papamichos-Chronakis *et al*, 2002) (Fig 4E). In accord, the differentially expressed genes in the evolved strains are enriched (8 out of 13) for transcription factor binding sites (TFBs) interacting with at least one of these three proteins (Fig EV5). This provides a mechanistic explanation for the increased flux through otherwise glucose-repressed reactions (e.g. glyoxylate shunt) in the evolved strains.

Our results show that integration of model-based strain design, adaptive laboratory evolution and multi-omics data can improve metabolic models even in well-studied organisms such as *S. cerevisiae*. While our study involved native metabolic products and relatively simple genetic designs, it provides a proof-of-concept for moving away from the one product–focussed strain design and warrants further studies with more complex metabolic engineering designs. In conclusion, the presented model-guided workflow, integrated with adaptive laboratory evolution and multi-omic characterization, holds potential for accelerated strain development, especially through automated engineering platforms such as bio-foundries.

# Materials and Methods

## Reagents and Tools table

| Reagent/Resource | Reference or source | Identifier or catalog number |
|---|---|---|
| ***Saccharomyces cerevisiae* strains** | | |
| CEN.PK119 (*MATa/MATalfa URA3/ura3-52 MAL2-8c/MAL2-8c SUC2/SUC2*) | Entian and Kötter (2007) | N/A |
| CEN.PK113-7D (*MATa URA3 MAL2-8c SUC2*) | Entian and Kötter (2007) | N/A |
| Engineered strains | This study | Appendix Table S7 |
| **Recombinant DNA** | | |
| pUG6 | EUROSCARF | P30114 |
| pUG74 | EUROSCARF | P30670 |
| pUG75 | EUROSCARF | P30671 |
| pYPK0 | Pereira *et al* (2016a) | N/A |
| **Oligonucleotides** | | |
| PCR primers | This study | Appendix Table S8 |
| **Chemicals, Enzymes and other reagents** | | |
| Defined minimal medium (MD) | Verduyn *et al* (1992) | N/A |
| Difco™ Yeast Nitrogen base without amino acids | BD™ 291940 | Cat # 11753573 |
| Ribitol | Alfa Aeaar | Cat # A17894 |
| RNAeasy kit | Qiagen | Cat # 74104 |
| Turbo DNAse | ThermoFisher | Cat # AM2238 |
| NEBNext® Ultra™ II Directional RNA Library Preparation Kit | New England Biolabs (NEB) | Cat # E7760S |
| NEBNext Poly(A) mRNA Magnetic Isolation Module | New England Biolabs (NEB) | Cat # E7490L |
| RapiGest SF Surfactants | Waters | Cat # 186001861 |
| Benzonase | Merck | Cat # 101654 |
| TMT10plex™ Isobaric Label Reagent | ThermoFisher | Cat # 90110 |
| NEBNext DNA Ultra2 Library Preparation Kit | New England Biolabs (NEB) | Cat # E7103 |
| **Software** | | |
| CPLEX ILOG Optimizer | IBM | v12.5.0, v12.8.0 |
| OptFlux workbench | http://www.optflux.org/ Rocha *et al* (2010) | v3.2.8, v3.4.0 |
| MEW – Metabolic Engineering Workbench | https://github.com/MEWorkbench | v1.1.0 |
| JECoLi – Java Evolutionary Computational Library | https://github.com/ | V1.1.0 |
| R A Language for Data Analysis and Graphics | https://www.r-project.org | v3.6.1, v3.6.3 |
| Python | www.python.org | v3.6.9 |
| GCMS solution software | Shimadzu Corp. | N/A |
| IsobarQuant | Franken *et al* (2015) | N/A |
| Mascot | www.matrixscience.com | v2.2.07 |
| Cell Growth Quantifier software | https://aquila-biolabs.de/ Aquila Biolabs | N/A |
| **Other** | | |
| Illumina NextSeq 500 | Illumina | N/A |
| Shimadzu TQ8040 GC | Shimadzu Corp. | N/A |
| GeneVac EZ-2 plus evaporating system | SP Scientific | N/A |
| Qubit | Thermo Fisher Scientific | N/A |
| Liquid handling robot | Beckman i7 | N/A |
| Cell disruptor, Sonifier | Branson | N/A |
| Illumina MiSeq System | Illumina | N/A |

**Reagents and Tools table** (continued)

| Reagent/Resource | Reference or source | Identifier or catalog number |
|---|---|---|
| Agilent 1200 Infinity high-performance liquid chromatography | Agilent | N/A |
| Automated liquid handling system | Hamilton Robotics | N/A |
| 2100 BioAnalyzer | Agilent Technologies | N/A |
| Precellys Lysing Kit | Bertin-instruments | Cat # KT0361-1-004.2 |
| OASIS® HLB µElution Plate | Waters | Cat # 186001828BA |
| ACQUITY UPLC/UHPLC System | Waters | N/A |
| Refractive Index (RI-2414) detector | Waters | N/A |
| ACQUITY UPLC PDA Detector | Waters | N/A |
| Rezex ROA-Organic Acid H+ (8%) column | Phenomenex | N/A |
| Aminex HPX-87H | Bio-Rad | N/A |
| ZB-50 capillary column | Phenomenex | Cat # 7HG-G004-11 |
| DASGIP® Parallel Bioreactor System for Microbial Applications | Eppendorf | N/A |
| Ultrospec® 2100 UV-Vis spectrophometer | Biochrom, Harvard Bioscience | N/A |
| Cell Growth Quantifier (CGQ) | https://aquila-biolabs.de/ Aquila Biolabs | N/A |
| MSM 400 micromanipulator | Singer Instruments | N/A |

## Methods and Protocols

### Metabolic models

The genome-scale metabolic model of yeast, iMM904, was used for the generation of results in the optimization and chassis analysis stages (Mo *et al*, 2009). The yeast models iND750 and Yeast 6 (Duarte *et al*, 2004; Heavner *et al*, 2013) were used for validation and fitness analysis. The iMM904 yeast model was modified in the following manner, to accommodate: (i) the addition of two mitochondrial 2-oxoglutarate transporters and a NAD-dependent cytosolic acetaldehyde dehydrogenase and (ii) the reversibility of three mitochondrial transporters (R_ASPt2m, R_OAAt2m and R_MALtm) were changed to prevent their activity in the direction from the mitochondria to the cytosol and the stoichiometry of the sirohydrochlorin dehydrogenase was corrected (Pereira *et al*, 2016b). The model was also modified to include a pyrophosphate transporter in the peroxisome, which was found to be essential to allow the activity of the glyoxylate cycle. After these modifications, the iMM904 model consists of 1,417 reactions, 1,064 internal metabolites and 904 genes. Modifications were also performed in the Yeast 6 model (Pereira *et al*, 2016a), including the update of several reactions according to the recent consensus model (Yeast 7.11) (Appendix Table S1).

### Selection of gene-deletion candidates for experimental verification

In the optimization stage, the improved model described above was used to filter the candidates for gene deletion using the following rules. First, essential or nearly essential reactions, determined as reactions that when excluded render biomass growth below 1% of the wild-type, were disregarded as potential targets. Next, all non-gene associated metabolic reactions, that either occur spontaneously, or due to lack of annotation, or do not possess any gene association, were also removed. Reactions that were unable to carry flux, for the tested environmental conditions, such as drains and transport reactions were also excluded. Finally, coupled reactions, co-sets or equivalent reactions, were grouped and only one of them was considered (Feist *et al*, 2010). After these strategies were employed, a total of 197 reactions were considered as possible deletion targets for the optimization stage.

### Nutrient uptake/secretion constraints

The case study was prepared to simulate fully aerobic growth using glucose as the sole carbon source. The glucose uptake flux was limited to a maximum of 1.15 mmol/gDW/h, and the oxygen uptake flux was unrestricted.

### Flux phenotype prediction

Phenotypic behaviour can be predicted using a number of constraint-based approaches over the information kept in metabolic models. In this work, several phenotype prediction methods were employed. Parsimonious enzyme usage Flux Balance Analysis (pFBA) (Lewis *et al*, 2010) was implemented as described in Carreira *et al* (2014) and used to evaluate all the candidate solutions during the optimization stage. Next, in the robustness analysis stage, Minimization of Metabolic Activity (MOMA) (Segrè *et al*, 2002), a linear variant of MOMA (lMOMA) (Lewis *et al*, 2012), was employed to evaluate the fitness of the solutions under alternative phenotype prediction methods. Finally, Flux Variability Analysis (FVA) (Zomorrodi *et al*, 2012) was used to check whether a given flux can vary in optimal FBA solutions, by setting a constraint that requires the biomass flux to be equal to its optimal value, and assess the robustness of a flux distribution regarding its production capability of the target compounds.

### Simulations of the contribution of a gene knockout to the production of the target compounds

All simulations presented in Appendix Fig S1 and Appendix Tables S2–S4 were performed in Optflux v3.2.8 using the modified version of the iMM904 model and the pFBA simulation method and CPLEX ILOG solver, with a glucose uptake rate of 1.15 mmol/gDW/h. And the simulations presented in Appendix Table S5 were performed in Optflux v3.2.8 using LMOMA simulation method and CPLEX ILOG solver, with a glucose uptake rate of 1.15 mmol/gDW/h.

### Platform strain design framework I: Optimization stage

The first step in the developed yeast chassis-strain design platform is the optimization stage. In this step, the gene deletion solutions for the improved production of each target compound are identified. For each of the target product (malic, succinic and fumaric acid), 10 optimization runs were executed. A metaheuristic approach based on the OptGene method was applied (Patil *et al*, 2005; Rocha *et al*, 2008). The employed heuristic is based on the Strength Pareto Evolutionary Algorithm 2 (SPEA2) (Zitzler *et al*, 2001; Rocha *et al*, 2010), readily available in the OptFlux workbench (Rocha *et al*, 2010). Being a multi-objective approach, the SPEA2 algorithm was configured to search for strain designs maximizing two conflicting objectives: (i) maximization of biomass and (ii) maximization of the target product. The optimization algorithm includes a solution archive manager that was configured to keep the 100 best solutions achieved during its execution. After 10 optimization runs for each target product, all the solutions were merged into a single solution set, discarding possible repetitions. The algorithm was allowed to generate solutions of variable sizes up to 20 deletions and, in each optimization run, a total of 100,000 function evaluations were allowed. A simplification process was applied where the deletions not directly contributing towards any of the objective functions were also ignored. Subsequently, the reaction-based solutions were converted to their gene-based equivalents, considering the GPR information available in the model. Finally, for each target product, every solution was evaluated against a set of metrics intended to support the chassis analysis stage, including:

- pFBA predicted growth and substrate uptake;
- FVA predicted range of variability of the target compound production;
- Biomass-Product Coupled Yield (BPCY) (Patil *et al*, 2005);
- Carbon Yield (CYIELD) (Equation 1), the yield of target production to substrate consumption, normalized by the carbon content of the respective compounds:

$$CYIELD(V) = \frac{v_{product} \times C_{product}}{v_{substrate} \times C_{substrate}}, \qquad (1)$$

where $C_{product}$ and $C_{substrate}$ correspond to the carbon content of a molecule of the product and substrate compounds, respectively;

- Cost of implementing a solution (COST), corresponding to the number of genes that compose a solution, a simplified metric to evaluate the time and financial cost of a wet laboratory implementation of such solution (assuming that a higher number of combined knockouts is increasingly expensive to implement and more likely to become experimentally unfeasible).

At the end of this stage, the results compiled for each of the target products were analysed together to find common chassis. All the optimizations were executed in a cluster environment composed of Intel Xeon processors with 2GB or RAM, each taking approximately 1 h to complete.

### Platform strain design framework II: chassis analysis

Solutions obtained in the optimization stage are clustered to support the chassis analysis. The framework resorts to the *heatmap.2* function from the *gplots R* package, which uses a hierarchical clustering procedure and produces an intuitive heat map. The distance matrix was calculated using the Euclidean distance metric, and cluster agglomeration was conducted via complete linkage. The procedure generates a heat map and dendrograms for both the rows (genes) and columns (target products), which are reordered by their mean distance values. Afterwards, a global score for each gene (gscore) (Equation 2) across all the target products was computed using the following scoring function:

$$gscore(g) = \left[ \sum_{t \in T} freq(g_t) \right]$$
$$\times \left[ \sum_{t \in T} x_{g,t}, where\, x_{g,t} = \left\{ \begin{array}{l} 1, \; if\, freq(g_t) > 0 \\ 1, \; if\, freq(g_t) = 0 \end{array} \right\} \right] \qquad (2)$$

where $g$ is a gene, $T = \{FA, SA, MA\}$ is the set of target products, $freq(g_t)$ is the frequency of gene $g$ in the target $t$ solution set, and $x_{g,t}$ is a binary variable for gene $g$ in target $t$ that takes the value 1 if $freq(g_t)$ is positive, or 0 otherwise. This equation translates into the sum of the frequencies for a gene across all the targets, being multiplied by the number of targets for which its frequency is different than zero, increasing the score of genes that occur for multiple targets.

In the third step, the genes were sorted according to their scores and the top-scoring genes were selected for chassis generation. For the design of yeast chassis for dicarboxylic acid production, the top 30% were used, computing all the possible combinations of those gene deletions up to size 5 generating multiple candidate chassis. Also, using the previously calculated results, all the solutions were organized in groups, where the criterion for each group is having a common chassis among them. In this stage, only solutions with at least 95% of the maximum identified carborn yield (CYIELD) for their respective target product were considered. Finally, a chassis score (cscore) (Equation 3) is also computed:

$$cscore(c) = \sum_{t \in T} \frac{\max[CYIELDS(S_{t,c})]}{\min[COSTS(S_{t,c})]}, \qquad (3)$$

where $c$ is the chassis, CYIELDS and COSTS are functions that return, respectively, a vector of CYIELD and COST values for a set of solutions, and $S_{t,c}$ is the set of solutions for target product $t$ containing chassis $c$. This scoring function promotes chassis that are valid for multiple target products and balances the score by diving the maximum CYIELD scoring solution by the minimum COST scoring solution in each set $S_{t,c}$. This score is only used to sort the chassis for analysis purposes, it is not used to exclude any chassis or solution. The final result is a list of chassis and respective solutions for each of the target products, sorted by their *cscore*.

### Yeast strain construction and maintenance

The *S. cerevisiae* strains used in this study, listed Appendix Table S7, have a CEN.PK genetic background (Entian & Kötter, 2007). Parental strain, CEN.PK119 (*URA3/ura3-52*), was derived from crossing strains CEN.PK113-5D and CEN.PK113-1A by zygote isolation using a SINGER micromanipulator. Deletion cassettes were obtained by the short flanking homology method (Wach *et al*, 1994) using pUG6 (Güldener *et al*, 1996), pUG74 or pUG75 as template (Hegemann & Heick, 2011) and with the primers listed in Appendix Table S8. Yeast strains were transformed as described (Schiestl & Gietz, 1989). Transformants were selected on solid YPD medium (1% yeast

extract, 2% peptone, 2% glucose, 1.5% agar; Formedium) supplemented with 200 mg/l G418 (Formedium), 100 mg/l nourseothricin (clonNat, Werner BioAgents) or 300 mg/l hygromycin B (Formedium). Single mutants were constructed by transforming the corresponding deletion cassette in the diploid strain CEN.PK119 (*URA3/ura3-52*). After tetrad analysis segregants of both mating types for *URA3* and *ura3-52* were isolated. Gene deletions of transformants and after tetrad analysis of haploid segregants were confirmed by diagnostic PCR using the primers listed in Table S8. Yeast strains with multiple deletions were generated by crossing, starting with the single deletion strains. The mating types of the strains constructed in this study were determined by PCR using whole yeast cells (Huxley *et al*, 1990). Plasmid containing *Δgdh1:: GDH2_URA3*MX4 cassette was assembled using TheYeastPathwayKit method (Pereira *et al*, 2016b). The strains were grown to stationary phase at 30°C and 180 rpm agitation in shake flask cultures on YPD medium (10 g/l of yeast extract, 20 g/l of peptone and 20 g/l of glucose). Stock cultures were prepared by adding glycerol to the culture broth (final concentration 30% v/v) and stored in sterile vials at −80°C. For YPD plates, 20 g/l of agar was included in the medium formulation. Pre-cultures were inoculated from a single colony of plated frozen stocks.

### Media and yeast cultivation

Yeast cells were grown in defined minimal medium (Verduyn *et al*, 1992) (MD) with 20 g/l of glucose (MD20), as carbon source, pH 5 on an orbital shaker set to 180 rpm at 30°C, unless indicated otherwise. Glycine auxotrophic strains were grown in the same medium supplemented with 500 mg/l of glycine, added by sterile filtration. Pre-cultures were prepared by cultivating a single colony from a selective YPD plate of each yeast strain in YPD medium until mid-exponential phase (~16 h). Pre-cultured cells were centrifuged, washed two times with distilled water, resuspended in MD and used for inoculation in MD with initial $OD_{600}$ of 0.1. These cultivations were performed in triplicates. Growth and multi-omics strain characterization were performed in MD20 with the procedure described above, whereas adaptive laboratory evolution was performed in MD media with high glucose concentration (100 g/l – MD100) in 50-ml shake flasks with 13 ml of MD100 medium at 30°C and constant agitation of 180 rpm.

### Adaptive laboratory evolution

Chassis_z and chassis-derived producing strains (Producer_F, Producer_Fz, Producer_Sz, Producer_Mz and Producer_FzG) were used in the adaptive laboratory evolution experiments. Adaptive laboratory evolution was performed by serial transfers of yeast cultures into fresh MD100 medium. Three independent parallel evolution cultures of each selected design were performed in shake flasks. Each independent culture was inoculated at an initial OD of 0.15 in fresh medium along the evolution experiment. Adaptive laboratory evolution experiment was conducted for 100 days, corresponding to an average of 24 serial passages. The first four consecutive passages were performed in medium supplemented with glycine, which was abruptly removed after the fourth passage. Thereafter, ALE was completed in MD100 without any amino acid supplementation. Culture stocks were frozen and stored at regular intervals throughout the process. The final evolved populations were plated in MD20+agar (Appendix Fig S2). Three isolated

colonies of each independent parallel population were picked for further physiological characterization, regarding their specific growth rate and production profile. All isolated clones from one of the parallel evolutions of Producer_F and Producer_Mz had high growth rates and lower lag-phases (Appendix Fig S3), comparable to wild-type strain. However, these isolates were no longer able to produce the respective target compound. The loss of production and improved fitness were not further investigated, neither the genotype of the isolates confirmed to rule out possible contaminations. The graphical pipeline of directed evolution followed by screening and the selection of the best growth-coupled producers employed in this study are depicted in Fig 2 and Appendix Figs S2 and S3.

### Controlled batch reactor cultivation

Batch fermentations were performed in YNB minimal medium containing 6.7 g/l of yeast nitrogen base without amino acids (Difco), supplemented with 50 g/l of glucose. Cells were pre-grown at 30°C and 180 rpm until mid-exponential phase was reached, in 250-ml shake flasks containing 50 ml of the same medium, and directly used for inoculation. Each fermenter was inoculated at an initial $OD_{600}$ of 0.1. The batch fermentations were performed in a DASGIP® Parallel Bioreactor System for Microbial Applications (Eppendorf) with 4 simultaneous culture vessels of 2 l with a working volume of 1 l. The temperature was maintained at 30°C, airflow rate was controlled at 1 VVM and constant pH of 5.5 was maintained by the automatic addition of 2 M NaOH. To ensure aeration, the dissolved oxygen was also monitored and kept above 30% of saturation by feedback control of the stirring speed from 400 rpm up to 800 rpm. The concentration of $O_2$ and $CO_2$, in the exhaust gas, was monitored by Bluesens off-gas analyzers. The batch cultures were sampled in regular intervals both in glucose and ethanol growth phases. Bioreactor cultivations were performed at least in duplicate.

### Biomass determination

Biomass concentration was monitored by measuring both dry cell weight (DCW) and optical density ($OD_{600}$) of the cultures. DCW was determined by filtering 5 ml of the fermentation broth and washed with 15 ml of distilled water, through a 0.22 µm pore filter from Millipore. Filters were pre-dried in a microwave oven at 150 W for 10 min, and the initial weight was measured using an analytical balance. After filtration, they were dried again in a microwave oven in the same conditions and stored in a desiccator before measuring the final weight. Optical density (OD) of the culture was estimated using a Ultrospec® 2100 UV-Vis spectrophotometer (Biochrom, Harvard Bioscience, Inc., USA) at a wavelength of 600 nm. In Fig 4, cell growth was followed using real-time measurements, the Cell Growth Quantifier (CGQ, Aquila Biolabs). The ODequivalent values were measured periodically during cultivation without perturbing cell culture.

### Sampling and quantification of fermentation products (shake flasks)

Samples were taken at precise times. After centrifugation (4,000 *g*, 10 min), supernatants were filtered using 0.2 µm PVDF—Poly (vinylidene fluoride)—syringe filter (Phenomenex, USA) and stored at −20°C until high-performance liquid chromatography (HPLC) and gas chromatography–mass spectrometry (GC-MS) analysis was

performed. Organic acids, including succinic acid, malic acid, fumaric acid, pyruvic acid and lactic acid were determined by GC-MS. Ribitol (Adonitol) (Alfa Aesar, UK) was added as an internal standard to the samples which were subsequently evaporated using a GeneVac EZ-2 plus evaporating system (SP Scientific). The dried extracts were derivatized with 50 ml of 20 mg/ml methoxyamine hydrochloride (Alfa Aesar, UK) solution in pyridine (Alfa Aesar, UK) for 90 min at 40°C, followed by reaction with 100 ml N-methyl-N-(trimethylsilyl)trifluoroacetamide (Alfa Aesar, UK) for approximately 12h at room temperature. Metabolites were measured using a Shimadzu TQ8040 GC-(triple quadrupole) MS system (Shimadzu Corp.). The gas chromatograph was equipped with a 30 m × 0.25 mm × 0.25 μm ZB-50 capillary column (7HG-G004-11, Phenomenex, USA). One microlitre of sample was injected in split mode at 250°C using helium as a carrier gas with a flow rate of 1 ml/min. GC oven temperature was held at 100°C for 4 min followed by an increase to 320°C with a rate of 10°C/min, and a final constant temperature period at 320°C for 11 min. The interface and the ion source were held at 280 and 230°C, respectively. The detector was operated both in scanning mode recording in the range of 50–600 $m/z$, as well as in Multiple Reaction Monitoring (MRM) mode for the specified metabolites. For peak annotation, the GCMS solution software (Shimadzu Corp.) was utilized. The metabolite identification was based on an in-house database with analytical standards utilized to define the retention time, the mass spectrum and the quantifying ion fragment for each specified metabolite. Metabolite quantification was performed by integrating the area under the curve (AUC) of the quantifying ion fragment of each metabolite (Supplementary Table 9) divided by the AUC of ribitol's quantifying ion ($m/z$ 319). For quantitative determination of succinate, malate and fumarate calibration curves were performed using known concentrations of target compounds.

Quantitative determination of extracellular compounds was also performed by ACQUITY UPLC/UHPLC System equipped with a Refractive Index (RI-2414) detector (Waters) and a ACQUITY UPLC PDA Detector. Glucose, succinic acid and ethanol were detected using RI detector, while acetate and fumaric acid were determined at 210 and 260 nm, respectively. The samples were analysed using a Rezex ROA-Organic Acid H+ (8%) column (Phenomenex, Aschaffenburg, Germany, kept at 65°C), where a solution of 0.5 mM of $H_2SO_4$ was used as the mobile phase with a flow rate of 0.5 ml/min. Calibration curves were performed with known concentrations of each metabolite, and target compounds were quantified by comparing the metabolite peak area in the sample with the peak area obtained in the sample.

### Quantification of fermentation products (bioreactors)

Samples for quantification of extracellular metabolites were obtained by rapidly taking 2 ml of broth, followed by immediate removal of the cells by filtration through PVDF syringe filters with a pore size of 0.22 μm (Millipore, USA). When $OD_{600} > 1$, the culture broth was first centrifuged (3,600 $g$, 8 min) and the supernatant was similarly filtered before being stored in HPLC vials at −20°C for further analysis. Subsequently, samples were analysed by HPLC (Jasco, Japan) model LC-NetII/ADC equipped with UV-2075 Plus and RI-2031 Plus detector. An Aminex HPX-87H column (Bio-Rad, kept at 65°C) was used and a solution of 0.01 M of $H_2SO_4$, with a flow rate of 0.5 ml/min, was used as the mobile phase. Quantitative

analysis of desired compounds was performed by comparison with a mixture of standards with known concentrations of each metabolite. Calibration curves were prepared using the peak areas of the RI detector for glucose, glycerol, acetate, ethanol and succinate, and of the UV absorbance for malate and fumarate.

### Genome sequencing and analysis of evolved and parental strains

Genomic DNA of population (at 5[th] passage) of Producer_Fz and isolated clones of evolved chassis-derived producing strains was extracted from cultures grown in YPD. At late-exponential stage, cells were pelleted and resuspended in 2 ml of TE buffer (0.1 M Tris and 0.1 M EDTA) supplemented with 1.5 U of Lyticase and incubated for 30 min at 37°C. Next, DNA was extracted and purified following the phenol-chloroform extraction protocol (Hoffman, 1997). DNA concentrations and quality were determined using Qubit (Thermo Fisher Scientific, USA). Equal amounts of DNA from all samples were used for library preparation, using the NEBNext DNA Ultra2 Library Preparation Kit (New England Biolabs). Library preparation was performed on an automated liquid handling system (Hamilton Robotics), and the quality of the library was tested on a 2100 BioAnalyzer (Agilent Technologies). Paired-end Illumina short read sequencing was performed of the whole genome DNA samples. The quality of the obtained reads was controlled using FastQC v. 0.10.1 (Andrews, 2010). Adapter removal, trimming of low-quality ends (both 3′ and 5′, quality cut-off 30) and short read filtering (minimum read length 31) were performed using cutadapt v.1.10 (Martin, 2011). The trimmed reads were aligned to *Saccharomyces cerevisiae* CEN.PK 113-7D genome assembly (Salazar *et al*, 2017) with the Burrows–Wheeler Aligner v.0.7.12 *bwa-mem* (Li & Durbin, 2009; preprint: Li, 2013) using default parameters. The alignments were processed (read groups added, sorted, reordered and indexed) and duplicate reads were marked using Picard Tools v.1.129 (http://broadinstitute.github.io/picard/). Single nucleotide variant (SNV) and insertion-deletion (indel) variant calling were performed with GATK4 v.4.1.0.0 (McKenna *et al*, 2010) *Haplotypecaller* in GVCF mode followed by joint genotyping using *GenotypeGVCF* (McKenna *et al*, 2010) using the *S. cerevisiae* CEN.PK113-7D genome assembly (Salazar *et al*, 2017) as the reference and default parameters. The called variants were filtered according to the GATK4 recommendations with the following thresholds for SNVs: QD < 2, FS > 60.0, MQ < 50.0, MQRankSum < −12.5, ReadPosRankSum < −8.0, GQ < 30 and DP < 5, and the following thresholds for indels: QD < 2, FS > 200, MQ < 50.0, ReadPosRankSum < −20.0, GQ < 30 and DP < 5, and the following thresholds for indels: QD < 2, FS > 200, MQ < 50.0, ReadPosRankSum < −20.0, GQ < 30 and DP < 5. Finally, the *de novo* variants were identified as the variants occurring only in independent evolved isolates and not in the non-evolved reference engineered strain. Variants were identified both in CDS locus as well as 500 bp upstream and downstream of gene. Structural variants (SVs) were called using Delly v.0.7.7 (Rausch *et al*, 2012). Called SVs were filtered against the non-evolved reference strain (Producer_Fz). The outputs were processed using BCFtools v.1.9 (Li, 2011) and svprops.

### Multi-omics sample preparation

Wild-type, evolved and parental chassis-derived producing strains were grown in MD20 (with parental strains supplemented with 500 mg/l of glycine). Samples were collected at mid-exponential

growth phase for transcriptomics, proteomics and extracellular metabolomics. For total RNA, 10 ml of culture broth was collected in a 50-ml Falcon® tube filled with ice and immediately centrifuged at 19,341 *g* for 2 min at 0°C. At the same time, for total protein, 10 ml of fermentation broth was transferred into ice-cold 15 ml Falcon® and immediately centrifuged at 10,000 *g* for 2 min at 0°C. After centrifugation, supernatant was discarded and cell pellet was washed once with ice-cold PBS buffer. After centrifugation, both cell pellets were snap-frozen in liquid nitrogen and kept at −80°C until extraction. For extracellular metabolomics, 5 ml of supernatant was collected and kept at −80°C until analysis. Sample preparation and quantification were performed by GS-MS as described.

### RNA-seq sample preparation and differential gene expression analysis

Total RNA was isolated with RNAeasy kit (Qiagen) following manufacturer recommendations. To remove DNA contamination, after extraction samples were digested with Turbo DNAse (Thermo Fisher) followed by RNA clean-up (RNAeasy kit, Qiagen). RNA library was prepared using the NEBNext® Ultra™ II Directional RNA Library Preparation Kit for Illumina: polyA transcripts capture. Barcoded stranded mRNA-seq libraries were prepared from ~200 ng total RNA samples using the NEBNext Poly(A) mRNA Magnetic Isolation Module and NEBNext Ultra II Directional RNA Library Prep Kit for Illumina (New England Biolabs (NEB), Ipswich, MA, USA) implemented on the liquid handling robot Beckman i7. Obtained libraries that passed the QC step were pooled in equimolar amounts; 2 pM solution of this pool was loaded on the Illumina sequencer NextSeq 500 and sequenced uni-directionally, generating ~500 million reads, each 85 bases long. Quality of the RNA-seq reads was assessed and summarized with Fastqc v.0.11.5 (Andrews, 2010). Next, cutadapt v.2.3 (Martin, 2011) was used for adapter trimming, to remove the standard lllumina TrueSeq Index adapters sequences. Quality read filtering and trimming was performed with FaQCs (Lo & Chain, 2014) v.2.08, with the following parameters: -q 20 -min_L 30 -n 3. Trimmed reads were then aligned to the reference genome of *S. cerevisiae* CEN.PK113-7D (EnsemblFungi: GCA_000269885) using STAR (Dobin *et al*, 2013) v.2.5.2a. On average, 95% of reads uniquely mapped to an annotated feature in the reference. Uniquely mapped reads were then used to generate the gene count tables with HTSeq (Anders *et al*, 2015) v.0.9.1. A total of 5430 genes with > 10 mapped reads were identified in Producer_F and Producer_Sz, while 5,433 were identified in Producer_Mz. Differential gene expression analysis, including multiple testing correction and independent filtering, was performed with Bioconductor package: DESeq2 (Love *et al*, 2014) v.1.12.0. False discovery rate (FDR) was calculated with fdrtool (Strimmer, 2008) v.1.2.15 using the raw *P*-values returned by DESeq2. Genes with a FDR < 0.1 were considered as significantly differentially expressed. Unless specified, all packages were used with default parameters. Biostatistical analysis was conducted with R v.3.6.1 (R Development Core Team).

### Protein sample preparation, sequencing and analysis

Frozen cell pellets were lysed using 0.1% RapiGest in 100 mM ammonium bicarbonate. Three cycles of sonication (1 cycle: 15 s sonication, 15 s on ice) (Cell disruptor, Sonifier, Branson) were applied to the lysate, followed by 15 min bead beating using Precellys Lysing Kit (KT0361-1-004.2). Cell lysate was transferred into a new tube after centrifugation (5 min, 5,000 *g*) and incubated at 80°C for 15 min. Benzonase (25 U, Merck) was added to the lysate and incubated for 30 min at 37°C. Cysteines were reduced using 10 mM of dithiothreitol (56°C, 30 min). The sample was cooled to 24°C and alkylated with 10 mM of iodacetamide (room temperature, in the dark, 30 min). Proteins were precipitated with TCA, and pellet was washed by acetone and dried. The proteins were digested in 50 mM HEPES (pH 8.5) using LysC (Wako) with an enzyme to protein ration 1:50 at 37°C for 4 h, followed by trypsin (Promega) with an enzyme to protein ratio 1:50 at 37°C overnight. TMT10plex™ Isobaric Label Reagent (Thermo Fisher) was added to the samples according the manufacturer's instructions. Labelled peptides were cleaned up using OASIS® HLB μElution Plate (Waters). Offline high pH reverse phase fractionation was performed using an Agilent 1200 Infinity high-performance liquid chromatography (HPLC) system, equipped with a Gemini C18 column (3 μm, 110 Å, 100 × 1.0 mm, Phenomenex) (Reichel *et al*, 2016). The solvent system consisted of 20 mM ammonium formate (pH 10.0) as mobile phase (A) and 100% acetonitrile as mobile phase (B). After fragmentation peptides were separated using the UltiMate 3000 RSLC nano LC system (Dionex) fitted with a trapping cartridge (μ-Precolumn C18 PepMap 100, 5 μm, 300 μm i.d. × 5 mm, 100 Å) and an analytical column (nanoEase™ M/Z HSS T3 column 75 μm × 250 mm C18, 1.8 μm, 100 Å, Waters). The outlet of the analytical column was coupled directly to a QExactive plus (Thermo) using the proxeon nanoflow source in positive ion mode. The peptides were introduced into the mass spectrometer (QExactive plus, Thermo Fisher) via a Pico-Tip Emitter 360 μm OD × 20 μm ID; 10 μm tip (New Objective) and a spray voltage of 2.3 kV was applied. The capillary temperature was set at 320°C. Full scan MS spectra with mass range 375–1,200 *m/z* were acquired in profile mode in the FT with resolution of 70,000. The peptide match algorithm was set to "preferred" and charge exclusion "unassigned", and charge states 1 and 5–8 were excluded. Isolation window was set to 1.0 and 100 *m/z* set as the fixed first mass. MS/MS data were acquired in profile mode (Strucko *et al*, 2018).

Acquired data were processed using IsobarQuant (Franken *et al*, 2015) and Mascot (v2.2.07). Searched against Uniprot *S. cerevisiae* CEN.PK113-7D proteome database. The following modifications were included into the search parameters: Carbamidomethyl (C) and TMT10 (K) (fixed modification), Acetyl (N-term), Oxidation (M) and TMT10 (N-term) (variable modifications). For the full scan (MS1), a mass error tolerance of 10 ppm and for MS/MS (MS2) spectra of 0.02 Da was set. Further parameters were set: Trypsin as protease with an allowance of maximum two missed cleavages: a minimum peptide length of seven amino acids; at least two unique peptides were required for a protein identification. The false discovery rate on peptide and protein level was set to 0.01.

Raw data of IsobarQuant were loaded into R. Only proteins that were quantified with two unique peptides were used for downstream analysis. The output data from IsobarQuant were cleaned for potential batch effects with limma (Ritchie *et al*, 2015) and subsequently normalized with vsn (variance stabilization) (Huber *et al*, 2002). Missing values were imputed with the impute function (method = "knn") from the MSNBase package (Gatto & Lilley, 2012). Under these conditions, a total of 3305 proteins were quantified and used to calculate differential protein abundances between tested strains. Differential abundance was performed with limma (Ritchie *et al*, 2015).

Proteins were classified as "hits" with a false discovery rate (fdr) <= 5% and a fold change of at least 200% and as "candidates" with fdr <= 20% and a fold change of at least 100%. The mass spectrometry proteomics data have been deposited to the ProteomeXchange Consortium via the PRIDE (Perez-Riverol *et al*, 2019) partner repository with the dataset identifier PXD020611.

### Integration of transcriptomics, proteomics and extracellular metabolomics data into flux balance analysis

Transcriptomics, proteomics and extracellular metabolomic data were integrated into flux balance analysis using MARGE function from *reframed* python package (https://doi.org/10.5281/zenodo.3478380). Flux balance analysis simulations of phenotype predictions were performed following MARGE standard parameter settings. The iMM904 yeast model initially used was transformed to include gene-protein-reaction (GPR) associations (Machado *et al*, 2016). Differences in extracellular metabolites abundances, gene expression levels and protein abundances of metabolic genes were calculated between the following: (i) parental chassis-derived producers and wild-type strains; and (ii) evolved and parental chassis-derived producing strains. Fold changes of differentially (*q*-value < 0.1) expressed metabolic genes, protein abundances and extracellular metabolite abundances between two conditions were used in the simulation tool. Relative growth rate differences between strains and wild-type (Fig 1D) were used to fit "growth_frac" parameter (growth_frac of wild-type = 1). Flux balance analysis was simulated using the differentially phenotypic changes imposed as lower/upper bounds in the flux of the respective reaction. The IBM ILOG CPLEX Optimizer (version 12.8.0) was used for solving the MILP problems. All simulations were conducted with Python 3.6.9.

### Updates to the yeast model for enabling flux through peroxisomal Mdh3p

The modified genome-scale metabolic model of yeast, iMM904, was further updated to include the missing glycerol-3-phosphate dehydrogenase reaction in the peroxisome identified in this study (Al-Saryi *et al*, 2017). The peroxisomal glycerol-3-phosphate dehydrogenase NAD-dependent reaction (R_G3PD1irp) was added to the model (associated to YDL022W gene). The model was also modified to include two new transport reactions cytosol-to-peroxisome of glycerol-3-phosphate (R_GLYC3Ptx) and peroxisome-to-cytosol of dihydroxyacetone phosphate (R_DHAPtx). After these modifications, the updated iMM904 model consists of 1,420 reactions, 1,066 internal metabolites and 904 genes.

## Data availability

The data sets and computer code produced in this study are available in the following databases:

- Modelling computer scripts and updated models: https://github.com/silicolife/yeastchassis/ (updated models under "data/models" folder).
- Genome sequencing data: ENA database with the identifier PRJEB41109 (https://www.ebi.ac.uk/ena/browser/view/PRJEB41109).

- RNA-Seq data: ArrayExpress database at EMBL-EBI under accession number E-MTAB-9499 (https://www.ebi.ac.uk/arrayexpress/experiments/E-MTAB-9499/).
- Mass spectrometry proteomics data: ProteomeXchange Consortium under accession number PXD020611 (via PRIDE (Perez-Riverol *et al*, 2019) partner repository). (https://www.ebi.ac.uk/pride/archive/projects/PXD020611).
- Extracellular metabolomics data: MetaboLights database at EMBL-EBI under the identifier MTBLS2007. (https://www.ebi.ac.uk/metabolights/MTBLS2007).

**Expanded View** for this article is available online.

## Acknowledgements

We would like to acknowledge the support of R. Mattel and F. Stein from the Proteomics Core Facility and the Genomics Core Facility at the European Molecular Biology Laboratory (EMBL Heidelberg, Germany). This study was supported by national funds through FCT/MCTES (Portugal, Ref. ERA-IB-2/0003/2013) and BMBF (Germany, Grant number: 031A343A, Ref. ERA-IB-2/0003/2013). The Portuguese Foundation for Science and Technology (FCT) supported HL through grant ref. PD/BD/52336/2013. FCT also supported this study under the scope of the strategic funding of UID/BIO/04469/2013 unit and COMPETE 2020 (POCI-01-0145-FEDER-006684) and through the Project RECI/BBB-EBI/0179/2012 (FCOMP-01-0124-FEDER-027462). Open Access funding enabled and organized by Projekt DEAL.

## Author contributions

Research design: FP, HL, IR, and KRP; Research: FP, HL, PM, BM, JN, DK, EK, MR, and PK; Data analysis: FP, HL, and PJ; Manuscript writing: FP, HL, and KRP; Final manuscript and final manuscript comments: All authors.

## Conflict of interest

The authors declare that they have no conflict of interest.

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
