## [Review Process File · Molecular Systems Biology]

Model-guided development of an evolutionarily stable yeast chassis

Filipa Pereira, Helder Lopes, Paulo Maia, Britta Meyer, Justyna Nocon, Paula Jouhten, Dimitrios Konstantinidis, Eleni Kafkia, Miguel Rocha, Peter Koetter, Isabel Rocha, and Kiran Patil
DOI: [10.15252/msb.202110253](https://doi.org/10.15252/msb.202110253)

Corresponding authors: Kiran Patil (kp533@cam.ac.uk) , Isabel Rocha (irocha@unl.pt)

Review Timeline:

Submission Date:	27th Jan 21
Editorial Decision:	15th Mar 21
Revision Received:	6th May 21
Editorial Decision:	9th Jun 21
Revision Received:	25th Jun 21
Accepted:	28th Jun 21

Editor: Maria Polychronidou

Transaction Report:

Manuscript Number: MSB-2021-10253, Model-guided development of an evolutionarily stable yeast chassis

Thank you again for submitting your work to Molecular Systems Biology. We have now heard back from the three referees who agreed to evaluate your study. Overall, the reviewers acknowledge that the study seems relevant for the field. They raise however a series of concerns, which we would ask you to address in a revision.

I think that the recommendations of the reviewers are rather clear and I therefore do not see the need to repeat the points listed below. As reviewer #3 mentions, providing some further evidence for the applicability of the approach to more complex designs would significantly enhance the impact of the study. All issues raised by the reviewers need to be satisfactorily addressed. Please contact me in case you would like to discuss in further detail any of the issues raised.

As a side note and following up on the recommendation of reviewer #2 we would encourage you to cite primary research papers in the Introduction instead of Reviews, whenever possible.

On a more editorial level, we would ask you to address the following points.

Reviewer #1:

This work uses a variety of omics methods to validate model-predicted genetic mutations in *Saccharomyces cerevisiae* for enhanced diacid accumulation. Using the metabolic model of *S. cerevisiae*, the authors were able to identify knockouts that enhance the secretion of the TCA cycle intermediate diacids: succinate, fumarate and malate. This provides strong support for the utility of model-guided host design. The presentation of the manuscript is clear and only minor points are listed below.

1. The authors identified a hyper-mutator phenotype in the strains lacking Fum1 while producing low concentrations of fumarate. This is unsurprising due to Fum1's role in the DNA damage response: <https://journals.plos.org/plosbiology/article?id=10.1371/journal.pbio.1000328>

2. Using the "adaptive laboratory evolution" strategy, the production of succinic acid and fumaric acid was increased, but not for malic acid. Can the author explain why the same approach isn't applicable for the next immediate downstream product, malic acid?

3. P11 L10: "Intrigued by these results..." this is a run-on sentence

Reviewer #2:

In the article entitled "Model-guided development of an evolutionarily stable yeast chassis" by Pereira et al. a metabolic engineering approach guided by metabolic model predictions is developed. The model predictions are validated with 3 different but related product, succinate, malate and fumarate. The in silico predictions are first experimentally validated by gene knock out and then, the strains were subject to adaptive laboratory evolution in order to unlock their full biosynthetic potential. In addition, the observed phenotypes were fully characterised using multiomics. The data has been made available in the relevant databases.

The work is nicely written and anticipate many of my questions with supporting experiments shown in supplementary materials.

I have just a few comments for the authors:

-The use of references in the introduction seems a bit strange. Some times, for simple sentences like "S. cerevisiae is a well-studied host" there are too many references that seem a bit randomly selected. Perhaps just citing a recent review is enough. This happens several times in the introduction.

-This work uses a combination of the iMM904 model and Yeast 7.11 and the model is further improved according to the results of the work (Adding missing reactions). It would be nice if the authors add as supplementary information the improved models as XML file.

-The following aspect of the method seems key to get the observed results: "To limit the number of mutations, a chassis score was then assigned to each solution based on the maximum carbon yield divided by the number of gene deletions that composes the solution". However, this seems a bit arbitrary and I am wondering the influence that selecting a different score would have in the results. for example, what is gene knock outs are limited to 5 or 6, are the predictions still the same? or much higher yields can be obtained? What is the rationale behind this score and was it selected based on any modeling evidence? nowadays is very easy to make a high number of knock outs in S. cerevisiae.

-While the authors found hundreds of genes with different expression level in the multiomic approach, the discussion seem to focus strongly in the enzymes shown in figure 3. Have the authors studied more generally other modifications? are all of them in metabolic genes? what about the others?

-For the genome sequencing of the evolved strains, single cells were isolated before sequencing, would they expect similar results if other colonies had been selected? did they verify that the selected one has a representative behaviour of the evolved community?

-page 9 line 35. I miss further information on how the multi-omic results were used to constrain the model.

-page 13, line 15. Did the author repeat the simulations with the updated model? what are the

differences when this was done?

-In the text, strains to produce one product have been evaluated for their capacity to produce such product, but have they been tested for their ability to produce the other products (e.g. the string modified to produce malate, is it producing some fumarate or succinate? or even other sink metabolites? is there any production of ethanol?)

Reviewer #3:

In this work, the authors develop designs for the growth-coupled production of succinate, fumarate, and malate in yeast utilizing metabolic modeling. The concept of model-driven design of growth-coupled strains to obtain an evolutionarily stable strain design has been around for almost two decades now, so there is nothing particularly new there. However, the authors execute a well-designed study and thoroughly investigate the phenotypes of their strains, mechanisms of adaptation, and failure modes.

The authors additionally introduce the work as chassis development, which is an appealing concept to shorten the strain development pipeline for a family of related products. However, the authors do not go far enough to show the utility of their chassis approach over developing independent strains. The 'chasses' in this case consist of two strains with 2 and 3 deletions each, which seems trivial to reproduce along the way to individual production strains and do not seem to have any additional engineering performed that would make them particularly suited to production. It is unfortunate that the authors chose such a simple design for their chassis, as more substantial chassis design and characterization, for example engineering additional properties that are likely to be broadly useful for production such as NADPH overabundance, would have made for a more interesting study.

The final conceptual issue is that the authors claim that evolutionary stability is achieved by this workflow. It is true that growth-coupled ALE strains exhibit evolutionarily stable production under a continued growth pressure. However, these ALE strains are rarely if ever the final production strain. Additional engineering is typically carried out to boost TRY to economically viable levels, which once again makes the strains vulnerable to evolutionary regression of production. The authors do not seem to make any progress on this issue relative to the status quo, neither do they perform any economic analysis that would suggest production metrics are sufficient in their current strains, so claiming to develop evolutionarily stable chassis strains (while technically true for the ALE strains) seems to be misleading.

In summary, the workflow is well executed but covers well-tread ground. I have a few additional specific comments on details that follow.

Additional specific comments

-The authors restrict complex designs in their computational design workflow, but it is for these complex designs where a chassis would seem to be the greatest utility, as a large number of deletions would only have to be performed once. It would be nice to see alternative options for chassis designs, as well as any differences in the resulting production strain performances.

-In constraint-based simulations of growth-coupled production, it is typical to show a 'production envelope' of production vs growth rate for the target compounds. This behavior can illustrate to what extent production is expected to depend on the growth rate. These envelopes would be helpful to show for their designs, as well as to compare to the observed growth and production rates.

-The 'debugging' of the failure of malate and fumarate producers was interesting - but it is not clear if the model predicted this or not. The NADPH depletion hypothesis was interesting but it is not clear why the model would not predict this issue - is it because of a lack of accounting for NADPH consumption mechanisms? Does adding an additional NADPH drain cause the model to predict that the design would fail as observed?

-Similarly, it is surprisingly that ALE did not improve malate production despite its prediction as a growth-coupled design - was this predicted by the model production envelope or not? It was not clear to me whether any model changes could be made that would fix this prediction. The authors suggested some changes in their discussion - did they try these changes to see if they fix the model prediction?

Reviewers comments:

Reviewer #1:

This work uses a variety of omics methods to validate model-predicted genetic mutations in *Saccharomyces cerevisiae* for enhanced diacid accumulation. Using the metabolic model of *S. cerevisiae*, the authors were able to identify knockouts that enhance the secretion of the TCA cycle intermediate diacids: succinate, fumarate and malate. This provides strong support for the utility of model-guided host design. The presentation of the manuscript is clear and only minor points are listed below.

> We thank the reviewer for their positive comments.

1. The authors identified a hyper-mutator phenotype in the strains lacking *Fum1* while producing low concentrations of fumarate. This is unsurprising due to *Fum1*'s role in the DNA damage

response: <https://journals.plos.org/plosbiology/article?id=10.1371/journal.pbio.1000328>

> We thank the reviewer for bringing the $\Delta fum1$ hyper-mutation phenotype to our notice. In our study, we observe that the hyper-mutation phenotype (>300 SNVs) manifested only in the fumarate producer strain deleted in *ZWF1* (*E_Producer_Fz: $\Delta ser3,33 \Delta zwf1 \Delta fum1$*). In the other two *FUM1* deleted strains, *E_Producer_F* ($\Delta ser3,33 \Delta fum1$) and *E_Producer_FzG* ($\Delta ser3,33 \Delta zwf1 \Delta fum1 \Delta gdh1::GDH2$), the number of mutations arising during the evolution are very low and comparable with the other evolved strains from the study (4 and 3, respectively). These observations indicate that the hyper-mutation phenotype is a synergistic response to both $\Delta fum1$'s role in the DNA damage process and the redox imbalance due to *ZWF1* deletion (which is rescued by the $\Delta gdh1::GDH2$ mutation). We have now updated the manuscript to note these observations (page 12, line 7) as bellow.

*“The deletion of FUM1 has been associated with increased sensitivity to DNA damage in yeast which can manifest into hyper-mutation phenotype (Yogev et al, 2010). In our evolution experiments, the hyper-mutation phenotype (>300 SNVs) is apparent only in one of the three of $\Delta fum1$ strains (*E_Producer_Fz: $\Delta ser3,33 \Delta zwf1 \Delta fum1$*). In the remaining $\Delta fum1$ strains, the one lacking *ZWF1* deletion (*E_Producer_F*) and the one engineered for improved NADPH availability (*Producer_FzG*), the number of identified mutations is low (<5 SNVs) (Fig EV4B). The hyper-mutation phenotype in the *E_Producer_Fz* yeast strain thus appears to be contingent on redox imbalance.”*

2. Using the "adaptive laboratory evolution" strategy, the production of succinic acid and fumaric acid was increased, but not for malic acid. Can the author explain why the same approach isn't applicable for the next immediate downstream product, malic acid?

> Our results suggest that during evolution the flux through *Mdh3p* (peroxisomal malate dehydrogenase) is increased converting the malate to oxaloacetate, which in turn is used to feed the TCA cycle. *Mdh3p* is up-regulated in both evolved and parental strains in comparison to wild-type, *Chassis_z*, and the other engineered

producer strains. Moreover, after evolution, Agx1p (converting glyoxylate to glycine) is down-regulated whereas Gly1p (converting threonine to glycine and acetaldehyde) is up-regulated. The evolved strain is thus likely producing glycine mainly through threonine aldolase (Gly1p), and the glyoxylate flux is partially deviated towards malate that in turn is converted to oxaloacetate by Mdh3p.

Peroxisomal malate dehydrogenase (Mdh3p), unlike the other two isoenzymes (Mdh1,2p), was not part of the initial gene deletion solutions identified by the model. Closer inspection and “de-bugging” of the model showed that even though the peroxisomal malate dehydrogenase activity was correctly included in the model, redox imbalance in peroxisome was blocking the flux through this reaction. *In vivo*, glycerol-3-phosphate dehydrogenase (encoded by *GPD1*) works together with malate dehydrogenase to balance redox in peroxisome. We identified that glycerol-3-phosphate dehydrogenase was not annotated in the model. By correcting the model, we were able to obtain flux through malate dehydrogenase and *MDH3* is now part of the target solutions set for malate production. Production envelopes for the target compounds using the updated IMM904 yeast model are presented at the end of this document (Figure R2 and Figure R3). As expected, the results for succinate and fumarate remain unchanged after model update (Figure R2). However, in the case of malate, growth-coupled production is only feasible with *MDH3* knockout in line with the model update (Figure R3).

We have expanded the Discussion (page 13, line 27) as bellow:

“With this update, flux balance analysis of the Producer_Mz strain genotype ($\Delta ser3,33\Delta zwf1\Delta mae1\Delta mdh1,2$) predicted flux through the reactions encoded by MDH3 and GDP1 and no malate secretion, as expected. Malate secretion phenotype was predicted when all three malate dehydrogenase isoenzymes (Mdh1,2,3) were deleted.”

3. P11 L10: "Intrigued by these results..." this is a run-on sentence

> Thank you for pointing this out. We have improved the sentence and the paragraph as below.

“Divergently, Gdh2p was up-regulated in the strain E_Producer_F (Fig 3). Engineering of GDH1 deletion and GDH2 overexpression has been shown to rescue phenotypes associated with low NADPH availability (Asadollahi et al, 2009; Brochado et al, 2010). We therefore investigated whether the fumarate producing strain deleted in ZWF1, Producer_Fz, decreased fumaric acid production during evolution due to low NADPH availability (Fig 4A).”

Reviewer #2:

In the article entitled "Model-guided development of an evolutionarily stable yeast chassis" by Pereira et al. a metabolic engineering approach guided by metabolic model predictions is developed. The model predictions are validated with 3 different but related product, succinate, malate and fumarate. The in silico predictions are first experimentally validated by gene knock out and then, the strains were subject to adaptive laboratory evolution in order to unlock their full biosynthetic potential. In addition, the observed phenotypes were fully

characterised using multiomics. The data has been made available in the relevant databases.

The work is nicely written and anticipate many of my questions with supporting experiments shown in supplementary materials.

> We thank the reviewer for positive assessment of the study and constructive comments for improvement.

I have just a few comments for the authors:

-The use of references in the introduction seems a bit strange. Some times, for simple sentences like "S. cerevisiae is a well-studied host" there are too many references that seem a bit randomly selected. Perhaps just citing a recent review is enough. This happens several times in the introduction.

> We agree and have accordingly revised the citations.

-This work uses a combination of the iMM904 model and Yeast 7.11 and the model is further improved according to the results of the work (Adding missing reactions). It would be nice if the authors add as supplementary information the improved models as XML file.

> We agree and have provided the XML models (both the one used for simulations, and a version corrected for peroxisomal redox balance based on our data) on GitHub: <https://github.com/silicolife/yeastchassis/tree/main/data/models>.

-The following aspect of the method seems key to get the observed results: "To limit the number of mutations, a chassis score was then assigned to each solution based on the maximum carbon yield divided by the number of gene deletions that composes the solution". However, this seems a bit arbitrary and I am wondering the influence that selecting a different score would have in the results. for example, what is gene knock outs are limited to 5 or 6, are the predictions still the same? or much higher yields can be obtained? What is the rationale behind this score and was it selected based on any modeling evidence? nowadays is very easy to make a high number of knock outs in S. cerevisiae.

> The chassis score is not imposed as a constraint or as a target in the simulations, but only used for ranking the solutions. It thus only ranks the identified solutions with higher predicted production (>95% of maximum identified carbon yield) by the number of gene deletions required. In the case of dicarboxylic acids, not much improvement is predicted beyond 3 gene deletions (Figure R1 presented at the end of this documented and Appendix Table S2-S4). For example, the predicted improvement for 4 gene knockouts solution over 3 gene knockouts is just 0.3% (Appendix Table S2-S4). We note that the chassis strain is only able to produce the target compound after additional product-specific mutation(s). Nevertheless, predicted target compound yield does not improve much with the increased number of knockouts (Figure R1 presented at the end of this documented).

We have updated the text to better explain the rationale behind chassis solutions as below (page 4, line 16):

"In brief, the algorithm searches for network modifications (in this case, one or more gene knockouts) such that the optimal flux distribution for biomass formation generates overflow of the target molecule (Burgard et al, 2003; Patil et al, 2005; Rocha et al, 2008) (Materials and Methods and Fig EV1). These solutions were then clustered to identify frequently occurring sets of gene targets common to all three products (Fig 1A). The solutions with >95% of maximum carbon yield for each target compound were ranked by the ratio between predicted carbon yield and the required number of gene (Chassis score) (Appendix Tables S2-S4). This ranking thus prioritizes solutions with higher carbon yield per network modification (Materials and Methods)."

-While the authors found hundreds of genes with different expression level in the multiomic approach, the discussion seem to focus strongly in the enzymes shown in figure 3. Have the authors studied more generally other modifications? are all of them in metabolic genes? what about the others?

> Figure 3 is aimed at providing an overview of the key changes in enzymes/pathways closely linked to the target compounds. Since the primary goal of this work is to evaluate the predictive capabilities of genome-scale metabolic models, we have focused the presentation of the multi-omics results on metabolic genes. Nevertheless, the complete datasets (i.e. including non-metabolic genes), were also used for the PCA analysis (Fig EV4), for the GO:term analysis (Appendix Fig S4), and for the analysis of transcription factor binding sites (Fig EV5). All the genomic, transcriptomic and proteomics data sets are publicly available (Data Availability Section) as well as in the source data file for Fig. EV4 ('SourceDataForFigureEV4.xlsx').

-For the genome sequencing of the evolved strains, single cells were isolated before sequencing, would they expect similar results if other colonies had been selected?

> We observed limited commonalities between the genetic changes in different evolved strains (Figure EV4B). In comparison, there is very good concordance between the predicted and the observed changes at gene/protein expression levels. As around half of the genetic changes concern regulatory genes, it is likely that the evolved populations are genetically heterogenous, yet convergent at the metabolic flux level.

did they verify that the selected one has a representative behaviour of the evolved community?

> After evolution experiment, the cell populations were plated in solid media and 3 single colonies (with different sizes: Small, Medium and Large) were isolated and characterized per population. Overall, the three independent isolated colonies (S,M and L) exhibited similar phenotype, at growth and production level, within each population (1, 2 or 3). One population for each of the target products was also characterized and the phenotypes of the isolated colonies resembled well that of the population. For further detailed characterization (i.e. compound production and multi-omics analysis), the colony with the highest production was selected. These results

are presented in Appendix Figure S3 (copied below) and in the Source Data for Figure S3.

Appendix Figure S3. Screening of three isolated strains from evolved populations.

A,B. Growth profile (A) and succinate production for evolved Producer_Sz isolated strains and Population 3 (B).

C,D. Growth profile (C) and fumarate production for evolved Producer_F isolated strains and Population 3 (D).

E,F. Growth profile (E) and malate production for evolved Producer_Mz isolated strains and Population 2 (F).

Data information: Target compounds titer was evaluated at the end of growth curve (Max OD₆₀₀). The OD₆₀₀ (A, C and E) was measured during cultivation for the 9 isolated colonies, one population and the respective parental strain without or with glycine supplementation (^{gly}). Populations (1,2 or 3)

were plated in solid minimal media with 20 g L⁻¹ of glucose and three isolated colonies (named S, M or L) were selected for characterization.

-page 9 line 35. I miss further information on how the multi-omic results were used to constrain the model.

> We have now revised the manuscript text (page 10, line 6) and methods section to make this clearer.

“Next, we integrated the multi-omics data with flux balance analysis to refine flux phenotype prediction. For this, we used relative changes in transcript and protein abundances between two conditions as constraints on the corresponding reaction rate (Machado et al, 2016). In addition, changes in extracellular metabolite concentrations were used to constrain the uptake/secretion of the corresponding metabolites. A flux balance solution consistent with these constraints was obtained using linear programming. The flux phenotype of the parental strain was predicted by comparing the omics data of the engineered producer strains to those of the wild-type. Whereas, the evolved strains were compared to their respective parental strains (Materials and methods).”

-page 13, line 15. Did the author repeat the simulations with the updated model? what are the differences when this was done?

> This is a very good suggestion and the revised manuscript includes the simulations with the corrected model . In brief, the model was updated to include the peroxisomal NAD-dependent reaction catalysed by glycerol-3-phosphate dehydrogenase, together with the respective peroxisomal drains for the metabolites. In the simulations with the updated model, as expected, no malate secretion was predicted for the genotype implemented in vivo ($\Delta ser3,33 \Delta zwf1 \Delta mae1 \Delta mdh1,2$). Malate secretion was predicted only when all three Mdh catalysed reactions ($\Delta ser3,33 \Delta zwf1 \Delta mae1 \Delta mdh1,2 \Delta mdh3$) were deleted.

Discussion section was updated to include the new simulations (page13, line 25), as below.

“Based on these findings, peroxisomal metabolism was updated in the yeast metabolic model - iMM904 - to include glycerol-3-phosphate dehydrogenase reaction (Materials and Methods). With this update, flux balance analysis of the Producer_Mz strain genotype ($\Delta ser3,33 \Delta zwf1 \Delta mae1 \Delta mdh1,2$) predicted no malate secretion and no flux through the reactions encoded by MDH3 and GDP1, as expected. Malate secretion was predicted when all three malate dehydrogenase isoenzymes (Mdh1,2,3) were deleted.”

-In the text, strains to produce one product have been evaluated for their capacity to produce such product, but have they been tested for their ability to produce the other products (e.g. the string modified to produce malate, is it producing some fumarate or succinate? or even other sink metabolites? is there any production of ethanol?

> We indeed measured production of a large number of metabolites for all strains (e.g. Figure EV4D). The results for the common fermentation metabolites (e.g.

ethanol, acetate, lactate, malate, succinate, fumarate, pyruvate and glucose) are now provided in Appendix Table S11.

At the end of fermentation, glucose was fully consumed in all evolved producer strains. Ethanol was produced, as expected, by all strains, although evolved strains showed a slight decrease in ethanol yield. In the evolved succinate and malate producer strains – deleted in *ZWF1* – acetate yield increased, compared to the respective parental strains, likely to offset the lower levels of cytosolic NADPH through aldolase dehydrogenase reaction. In contrast, acetate yield decreased in evolved fumarate producer strain.

Dicarboxylic acid production was also measured in all producer strains, before and after evolution. As expected, strain engineered to produce succinate does not secrete appreciable amounts of fumarate and malate. In the malate producer strain, while fumarate secretion increased compared to Chassis and WT, it still remains very low (~0.04 g/L). Further, both fumarate and succinate secretion decreased after evolution this strain. The fumarate producer strain secreted both fumarate (0.253 g/L) and succinate (0.189 g/L, ~60% less than the obtained in succinate producer strain). Nevertheless, after evolution, fumarate production increased by 70% (0.352 g/L) while the succinate production decreased by 60% (0.108 g/L).

Discussion section was updated to include the observed by-product formation (page 8, line 21), as below.

" Production of non-target dicarboxylic acids (i.e. fumarate and malate) was not observed in succinate producer strains (Appendix Table S11). On the other hand, fumarate producing Producer_F strain also secreted succinate, albeit in much less amounts than the succinate producing Producer_Sz strain. Nevertheless, after evolution, fumarate production in this strain increased by 70%, while the succinate production decreased by 60%. The robustness of growth-coupled malate production was only predicted when fumarate drain was impaired (Appendix Fig S1G). In vivo, this strain secreted fumarate in low amounts (~0.04 g/L), which further dropped after evolution (Appendix Table S11)."

Reviewer #3:

In this work, the authors develop designs for the growth-coupled production of succinate, fumarate, and malate in yeast utilizing metabolic modeling. The concept of model-driven design of growth-coupled strains to obtain an evolutionarily stable strain design has been around for almost two decades now, so there is nothing particularly new there. However, the authors execute a well-designed study and thoroughly investigate the phenotypes of their strains, mechanisms of adaptation, and failure modes.

> We thank the reviewer for their appreciation of our thorough study design and execution.

The authors additionally introduce the work as chassis development, which is an appealing concept to shorten the strain development pipeline for a family of related products. However, the authors do not go far enough to show the utility of their chassis approach over developing independent strains. The 'chassis' in this case

consist of two strains with 2 and 3 deletions each, which seems trivial to reproduce along the way to individual production strains and do not seem to have any additional engineering performed that would make them particularly suited to production. It is unfortunate that the authors chose such a simple design for their chassis, as more substantial chassis design and characterization, for example engineering additional properties that are likely to be broadly useful for production such as NADPH overabundance, would have made for a more interesting study.

> We cannot agree more; a model-designed chassis with a broader scope is indeed highly desirable. Yet, the progress in this field is hampered by the shortage of studies that go the full design (esp. model-driven design) – build – test – improve cycle. Our study does this, and, thus, we believe it is an essential step towards building more complex chassis.

The final conceptual issue is that the authors claim that evolutionary stability is achieved by this workflow. It is true that growth-coupled ALE strains exhibit evolutionarily stable production under a continued growth pressure. However, these ALE strains are rarely if ever the final production strain. Additional engineering is typically carried out to boost TRY to economically viable levels, which once again makes the strains vulnerable to evolutionary regression of production. The authors do not seem to make any progress on this issue relative to the status quo, neither do they perform any economic analysis that would suggest production metrics are sufficient in their current strains, so claiming to develop evolutionarily stable chassis strains (while technically true for the ALE strains) seems to be misleading.

> The goal of this study was to challenge the models for their predictive capability and to investigate in-depth how well the model predictions are manifested at molecular level. Building a production strain for industrial application is out of scope for this work. In fact, many excellent examples of high TRY strains are already published (Burgard *et al.*, 2016); yet, one can note that few of these if any at all are based in model predicted designs.

Reference: Burgard A, Burk MJ, Osterhout R, Van Dien S, Yim H. Development of a commercial scale process for production of 1,4-butanediol from sugar. *Current Opinion in Biotechnology*. 2016 42:118-125.

In summary, the workflow is well executed but covers well-tread ground. I have a few additional specific comments on details that follow.

Additional specific comments

-The authors restrict complex designs in their computational design workflow, but it is for these complex designs where a chassis would seem to be the greatest utility, as a large number of deletions would only have to be performed once. It would be nice to see alternative options for chassis designs, as well as any differences in the resulting production strain performances.

> We believe that this topic might be insufficiently clear in the results section, therefore we have updated the text to better explain the rationale behind chassis solutions (page 4, line18). In the optimization stage, the algorithm was allowed to

generate solutions of up to 20 deletions and, in each optimization run, a total of 100,000 function evaluations were allowed. From this pool the solutions with high predicted production (>95% of maximum identified carbon yield) were sorted by the number of gene deletions required to obtain such phenotype. For the production of dicarboxylic acids, we used a cut-off of 5 gene deletions for identifying the chassis design, however the selected chassis for experimental implementation was composed by 3 gene deletions. There is not much increase predicted beyond 3 gene deletions (Figure R1 at the end of this document, and Appendix Table S2-S4). For example, the predicted improvement for 4 gene knockouts solution over 3 gene knockouts is just 0.3% (Appendix Table S2-S4). We note that the chassis strain is only able to produce the target compound after additional product-specific mutation(s). Nevertheless, predicted target compound yield does not improve much with the increased number of knockouts (Figure R1 presented at the end of this documented).

We have updated the text to better explain the rationale behind chassis solutions as below (page 4, line 16):

"In brief, the algorithm searches for network modifications (in this case, one or more gene knockouts) such that the optimal flux distribution for biomass formation generates overflow of the target molecule (Burgard et al, 2003; Patil et al, 2005; Rocha et al, 2008) (Materials and Methods and Fig EV1). These solutions were then clustered to identify frequently occurring sets of gene targets common to all three products (Fig 1A). The solutions with >95% of maximum carbon yield for each target compound were ranked by the ratio between predicted carbon yield and the required number of gene (Chassis score) (Appendix Tables S2-S4). This ranking thus prioritizes solutions with higher carbon yield per network modification (Materials and Methods)."

-In constraint-based simulations of growth-coupled production, it is typical to show a 'production envelope' of production vs growth rate for the target compounds. This behavior can illustrate to what extent production is expected to depend on the growth rate. These envelopes would be helpful to show for their designs, as well as to compare to the observed growth and production rates.

> Thank you for this excellent suggestion. The 'production envelop' obtained for each target compound is now added in the Appendix Figure S1E-G (reproduced below).

Updated Appendix Figure S2. Predicted effect of gene deletion targets for C4-dicarboxilic acids production.

E Predicted 'Production envelop' for succinate production in the *in vivo* implemented solution ($\Delta ser3,33\Delta zwf1\Delta sdh3$)

F Predicted 'Production envelop' for fumarate production in the *in vivo* implemented solution ($\Delta ser3,33\Delta zwf1\Delta fum1$)

G Predicted 'Production envelop' for malate production in the *in vivo* implemented solution ($\Delta ser3,33\Delta zwf1\Delta mae1\Delta mdh1,2$) + impaired fumarate drain

E,F,G 'Production envelop' for succinate (E), Fumarate (F) and Malate (G) using the initial iMM904 yeast model.

-The 'debugging' of the failure of malate and fumarate producers was interesting - but it is not clear if the model predicted this or not. The NADPH depletion hypothesis was interesting but it is not clear why the model would not predict this issue - is it because of a lack of accounting for NADPH consumption mechanisms? Does adding

an additional NADPH drain cause the model to predict that the design would fail as observed?

> While the genome-scale metabolic models do account for the major production and depletion routes of NADPH, they are yet poorly suited to account for thermodynamic constraints on the use of reactions where NADH can be used as an alternative for NADPH. The models thus consider these two redox co-factors as equivalent, e.g. Gdh1 or Gdh2 routes, which are not equivalent *in vivo*. Further, NADPH depletion can be compensated *in silico* by transhydrogenation cycles but not *in vivo*. The omics analysis, on the other hand, did reveal the redox imbalance as we discuss in the manuscript.

-Similarly, it is surprisingly that ALE did not improve malate production despite its prediction as a growth-coupled design - was this predicted by the model production envelope or not? It was not clear to me whether any model changes could be made that would fix this prediction. The authors suggested some changes in their discussion - did they try these changes to see if they fix the model prediction?

> Our results suggest that during evolution the flux through Mdh3p (peroxisomal malate dehydrogenase) is increased converting the malate to oxaloacetate, which in turn is used to feed the TCA cycle. Mdh3p is up-regulated in both evolved and parental strains in comparison to wild-type, Chassis_z, and the other engineered producer strains. Moreover, after evolution, Agx1p (converting glyoxylate to glycine) is down-regulated whereas Gly1p (converting threonine to glycine and acetaldehyde) is up-regulated. The evolved strain is thus likely producing glycine mainly through threonine aldolase (Gly1p), and the glyoxylate flux is partially deviated towards malate that in turn is converted to oxaloacetate by Mdh3p. Peroxisomal malate dehydrogenase (Mdh3p), unlike the other two isoenzymes (Mdh1,2p), was not part of the initial gene deletion solutions identified by the model. Closer inspection and “de-bugging” of the model showed that even though the peroxisomal malate dehydrogenase activity was correctly included in the model, redox imbalance in peroxisome was blocking the flux through this reaction. *In vivo*, glycerol-3-phosphate dehydrogenase (encoded by *GPD1*) works together with malate dehydrogenase to balance redox in peroxisome. We identified that glycerol-3-phosphate dehydrogenase was not annotated in the model. By correcting the model, we were able to obtain flux through malate dehydrogenase and *MDH3* is now part of the target solutions set for malate production. Production envelopes for the target compounds using the updated IMM904 yeast model are presented at the end of this document (Figure R2 and Figure R3). As expected, the results for succinate and fumarate remain unchanged after model update (Figure R2). However, in the case of malate, growth-coupled production is only feasible with *MDH3* knockout in line with the model update (Figure R3).

We have expanded the Discussion (page 13, line 27), as bellow:

*“With this update, flux balance analysis of the Producer_Mz strain genotype ($\Delta ser3, 33\Delta zwf1\Delta mae1\Delta mdh1,2$) predicted predicted flux through the reactions encoded by *MDH3* and *GDP1* and no malate secretion, as expected. Malate*

secretion phenotype was predicted when all three malate dehydrogenase isoenzymes (*Mdh1,2,3*) were deleted.”

Figures:

Figure R1. Predicted carbon yield as a function of number of gene deletions.

Figure R2. ‘Production envelop’ for the predicted growth-coupled production of target compounds using the updated IMM904 yeast model.

- A** Predicted ‘Production envelop’ for succinate production in the *in vivo* implemented solution ($\Delta ser3,33\Delta zwf1\Delta sdh3$)
- B** Predicted ‘Production envelop’ for fumarate production in the *in vivo* implemented solution ($\Delta ser3,33\Delta zwf1\Delta fum1$)

Figure R3. 'Production envelop' for the predicted growth-coupled production of malate using the updated IMM904 yeast model with the disruption of fumarate drain.

C Simulation using the set of gene deletions implemented *in vivo*,

D Simulation with the deletion of *MDH3* included in the set of solutions implemented *in vivo* for malate production.

RE: MSB-2021-10253R, Model-guided development of an evolutionarily stable yeast chassis

Thank you for sending us your revised manuscript. We have now heard back from the two reviewers who were asked to evaluate your study. As you will see below, the reviewers are satisfied with the modifications made and are supportive of publication.

Before we can formally accept the study for publication we would ask you to address the editorial issues listed below.

Reviewer #2:

The authors have addressed all my concerns

Reviewer #3:

The authors satisfactorily answered my technical concerns. I don't have any new issues to raise.

The authors performed the requested editorial changes.

RE: MSB-2021-10253RR, Model-guided development of an evolutionarily stable yeast chassis

Thank you again for sending us your revised manuscript. We are now satisfied with the modifications made and I am pleased to inform you that your paper has been accepted for publication.

Corresponding Author Name: Kiran Raosaheb Patil

Manuscript Number: MSB-2021-10253